# ZeroMark: Towards Dataset Ownership Verification without Disclosing Watermarks

**Junfeng Guo**[1,*]**, Yiming Li**[2,*]**, Ruibo Chen**[1]**, Yihan Wu**[1]**, Chenxi Liu**[1]**, Heng Huang**[1]

[1]Department of Computer Science, Institute of Health Computing
University of Maryland College Park
[2]College of Computing and Data Science, Nanyang Technology University
{gjf2023,ruibo,yihanwu,chenxi,heng}@umd.edu; liyiming.tech@gmail.com

## Abstract

High-quality public datasets significantly prompt the prosperity of deep neural networks (DNNs). Currently, dataset ownership verification (DOV), which consists of dataset watermarking and ownership verification, is the only feasible solution to protect their copyright by preventing unauthorized use. In this paper, we revisit existing DOV methods and find that they all mainly focused on the first stage by designing different types of dataset watermarks and directly exploiting watermarked samples as the verification samples for ownership verification. As such, their success relies on an underlying assumption that verification is a *one-time* and *privacy-preserving* process, which does not necessarily hold in practice. To alleviate this problem, we propose *ZeroMark* to conduct ownership verification without disclosing dataset-specified watermarks. Our method is inspired by our empirical and theoretical findings of the intrinsic property of DNNs trained on the watermarked dataset. Specifically, ZeroMark first generates the closest boundary version of given benign samples and calculates their boundary gradients under the label-only black-box setting. After that, it examines whether the given suspicious method has been trained on the protected dataset by performing a hypothesis test, based on the cosine similarity measured on the boundary gradients and the watermark pattern. Extensive experiments on benchmark datasets verify the effectiveness of our ZeroMark and its resistance to potential adaptive attacks. The codes for reproducing our main experiments are publicly available at [GitHub](GitHub).

## 1 Introduction

Deep neural networks (DNNs) have demonstrated their strong ability in widespread applications, such as face recognition [1, 2, 3]. Currently, there are many (high-quality) public datasets, such as CIFAR [4] and ImageNet [5], that can be easily downloaded and used. Arguably, their availability is one of the key factors in the prosperity of DNNs, as developers can evaluate and improve their models upon them. In particular, these datasets are usually only freely available for non-commercial use since their collection and annotation are time-consuming and even expensive.

To the best of our knowledge, dataset ownership verification (DOV) [6, 7, 8, 9, 10, 11] is currently the only feasible solution for protecting the copyright of public datasets. Specifically, DOV consists of two main stages, including dataset watermarking and ownership verification. In the first stage, dataset owners will introduce some imperceptible watermarked samples to generate the released watermarked version of the original dataset, so that all models trained on it will have specific distinctive prediction behaviors on particular samples (*i.e.*, verification samples) while having normal

---

*The first two authors contributed equally to this work.

38th Conference on Neural Information Processing Systems (NeurIPS 2024).

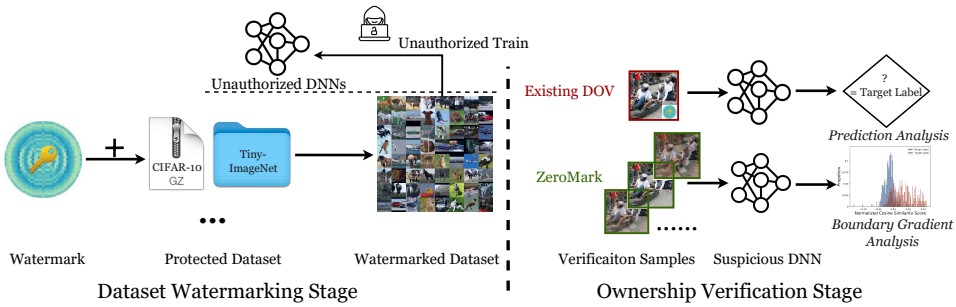

Figure 1: The overview of existing dataset ownership verification (DOV) methods and our Zero-Mark. In the verification phase, existing DOV approaches directly exploit watermarked samples for verification purposes. In contrast, ZeroMark queries the suspicious model with boundary samples without disclosing dataset-specified watermarks to safeguard the verification process.

behaviors on standard testing samples. In the second stage, given the API of a suspicious third-party deployed model, the dataset owners will detect whether it is trained on the protected dataset by examining its prediction behaviors on verification samples.

In this paper, we revisit existing DOV methods. We find that they all mainly focused on the first stage by designing different types of dataset watermarks, no matter whether their watermark is backdoor-based [6, 7, 8] or not [9, 11]. All of them directly exploited watermarked samples as the verification samples in their second stage. Accordingly, their success relies on an underlying assumption that verification is a *one-time* and *privacy-preserving* process. Otherwise, as the watermark pattern is leaked during the (first) verification process, the malicious dataset users can easily remove the watermark from the model trained on the stolen dataset or from the verification samples. However, this assumption does not necessarily hold true in practice since adversaries can always update their unauthorized models. As such, an intriguing and critical question arises: *Could we verify dataset ownership without disclosing dataset-specified watermarks to ensure a secure verification process*?

The answer to the aforementioned problem is positive. In this paper, we first delve into the intrinsic property of DNNs trained on the watermarked dataset. We empirically and theoretically demonstrate that the gradient of watermarked models calculated upon the closest samples located at the decision boundary (dubbed 'boundary gradient') has a similar direction to their corresponding watermark patterns, measured by their cosine similarity. Specifically, the distribution of cosine similarity of the watermarked model on the dataset-specified target class has significantly larger values than that of the remaining (benign) classes. In particular, samples located at the decision boundary (dubbed 'boundary samples') of watermarked DNNs contain limited information about the dataset-specified watermarks, measured by several metrics (*e.g.*, mutual information). Motivated by these intriguing findings, we propose to conduct dataset ownership verification with closest boundary samples instead of samples containing dataset-specified watermarks (as shown in Figure 1). We call this method as ZeroMark. Specifically, our ZeroMark has three main steps. In the first step, ZeroMark generates the closest boundary version of given benign samples. The second step calculates the boundary gradients of generated closest boundary samples based on the Monte Carlo method. Both steps are conducted under the label-only black-box verification setting, where dataset owners can only query the suspicious model with verification samples via API and get its predicted labels. In the third step, ZeroMark examines whether the given suspicious method has been trained on the protected dataset via a hypothesis test, based on the distribution of cosine similarity measured between the boundary gradients and the corresponding watermark pattern.

In conclusion, our main contributions are four-fold: **(1)** We revisit existing DOV methods and reveal their underlying assumption regarding the verification phase. It does not necessarily hold, hindering the protection of dataset copyright. **(2)** We empirically and theoretically discover an intrinsic property of watermarked DNNs regarding boundary samples (*i.e.*, those located at the decision boundary). **(3)** We propose a simple yet effective method (*i.e.*, ZeroMark) to verify dataset ownership without disclosing dataset-specified watermarks. **(4)** We conduct experiments on benchmark datasets, verifying the effectiveness of ZeroMark and its resistance to potential adaptive methods.

## 2 Related Work

### 2.1 Data Protection

**Classical Data Protection.** Data protection is a classical and significant research area, which aims to prevent unauthorized data usage or protect private data. Existing classical data protection con-

sists of three main categories, including **(1)** encryption, **(2)** digital watermarking, and **(3)** privacy protection. Specifically, encryption [12, 13, 14] encrypts the sensitive data so that only authorized users with a secret key can decrypt and use it. Digital watermarking [15, 16, 17] embeds an owner-specified pattern to the protected data as the watermark to claim ownership. Privacy protection focuses on preventing the leakage of sensitive information of the data in both empirical [18, 19, 20] and certified manners [21, 22, 23]. Unfortunately, existing classical approaches can not directly protect the copyright of open-source datasets since they either hinder the dataset accessibility (*e.g.*, encryption) or require the information of the training process of models trained on them (*e.g.*, digital watermarking and privacy protection) that will not be disclosed by authorized dataset users.

**Dataset Ownership Verification.** Dataset ownership verification (DOV) aims to verify whether a suspicious model is trained on the protected dataset. To the best of our knowledge, this is currently the only feasible method to protect the copyright of open-source datasets. Specifically, DOV intends to introduce specific prediction behaviors (towards verification samples) in models trained on the protected dataset while preserving their performance on benign testing samples. Dataset owners can verify ownership by examining whether the suspicious model has dataset-specified distinctive behaviors. Previous DOV methods exploit either backdoor attacks [6, 7, 8, 10] or others [9, 11] to watermark the original (unprotected) benign dataset. For example, recently, backdoor-based DOV [6, 8, 7] adopted poisoned-/clean-label backdoor attacks to watermark the protected dataset. Most recently, Guo *et al.* [9] adopted samples from the hardly-generalized domain as watermark samples without introducing any new security vulnerability. However, all existing dataset ownership verification (DOV) approaches [6, 8, 7, 9, 10, 11] mainly focus on designing watermarks with different properties (*e.g.*, harmless and stealthy) and directly exploit the watermarked samples for verification. The security study of their verification stage remains blank and is worth further exploration.

## 2.2 Secure Machine Learning Inference

Currently, there are also a few works to safeguard the inference process of models. In the context of machine learning, secure inference is a two-party cryptographic protocol applied in the inference phase of machine learning models [24, 25, 26]. The server learns nothing about clients' input, while a client learns nothing about the server's machine learning model but can only get the results. Technically, it is implemented by having the server and client involved in a specific protocol and running the encrypted model over the encrypted input through cryptographic techniques such as homomorphic encryption [27] and secret sharing [28]. However, secure inference requires both the client and server to encrypt input data and adapt the machine learning model's operations accordingly through cryptographic mechanisms [27, 28]. As such, it is infeasible to protect the verification process of DOV methods since suspicious third-party models may not support these protocols.

# 3 The Property of Models Trained on the Watermarked Dataset

## 3.1 Preliminaries

**The Main Pipeline of Existing DOV Methods.** Let $\mathcal{D} = \{(\boldsymbol{x}_i, y_i)\}_{i=1}^N$ denotes the original training dataset. In context of image classification task with $K$-classes, *i.e.*, $\boldsymbol{x}_i \in \mathcal{X} = [0,1]^{C \times W \times H}$ represents the image with $y_i \in \mathcal{Y} = \{1, \cdots, K\}$ as its label. In the first stage of DOV (*i.e.*, dataset watermarking), the dataset owner will embed watermarks to the original dataset to generate its watermarked version (*i.e.*, $\mathcal{D}_w$). Particularly, $\mathcal{D}_w = \mathcal{D}_m \cup \mathcal{D}_b$, where $\mathcal{D}_m$ represents the watermarked version of samples from a small selected subset $\mathcal{D}_s$ of $\mathcal{D}$ (*i.e.*, $\mathcal{D}_s \subset \mathcal{D}$) and $\mathcal{D}_b$ contains remaining benign samples (*i.e.*, $\mathcal{D}_b = \mathcal{D} - \mathcal{D}_s$). The $\mathcal{D}_m$ is generated by the dataset-specified image generator $G_x : \mathcal{X} \to \mathcal{X}$ and the label generator $G_y : \mathcal{Y} \to \mathcal{Y}$, *i.e.*, $\mathcal{D}_m = \{(G_x(\boldsymbol{x}), G_y(y))|(\boldsymbol{x}, y) \in \mathcal{D}_s\}$. For example, $G_x = (\mathbf{1} - \boldsymbol{m}) \odot \boldsymbol{\Delta} + \boldsymbol{m} \odot \boldsymbol{x}$ and $G_y = y_t$ in BadNets-based DOV [29, 6], where $\boldsymbol{m} \in \{0,1\}^{C \times W \times H}$ is the trigger mask, $\boldsymbol{\delta} \in [0,1]^{C \times W \times H}$ is the trigger pattern, $\odot$ denotes the element-wise product, and $y_t$ is the target label. In particular, $\gamma \triangleq \frac{|\mathcal{D}_m|}{|\mathcal{D}_w|}$ is the *watermarking rate*. In the second phase (*i.e.*, ownership verification), for a suspicious model $C : \mathcal{X} \to \mathcal{Y}$ that may be trained on $\mathcal{D}_w$, the dataset owners will investigate whether it conducts unauthorized training by querying it with verification samples under the black-box setting. In general, the verification process of existing DOV is to directly uses watermarked sample $G_x(\boldsymbol{x})$ as verification samples to examine

whether $C(G_x(\boldsymbol{x})) = G_y(y)$. In contrast, our goal is to perform the verification process without disclosing the watermark samples $G_x(\boldsymbol{x})$ during the inference phase of the suspicious classifier $C$.

**Boundary Samples.** Let the logit margin of model $f : \mathcal{X} \to [0, 1]^K$ on the label $y$ is denoted by:

$$\phi_y(\boldsymbol{x}; \boldsymbol{w}) = f_y(\boldsymbol{x}; \boldsymbol{w}) - \max_{y' \neq y} f_{y'}(\boldsymbol{x}; \boldsymbol{w}). \tag{1}$$

It can be observed that $\boldsymbol{x}$ can be classified as $y$ by $f(\cdot; \boldsymbol{w})$ if and only if $\phi_y(\boldsymbol{x}; \boldsymbol{w}) \geq 0$. As such, the set for boundary samples of class $y$ can be denoted by $\mathcal{B}(y; \boldsymbol{w}) = \{x : \phi_y(\boldsymbol{x}; \boldsymbol{w}) = 0\}$.

### 3.2 Approach the Closest Boundary Sample

To obtain the boundary samples, we can easily use a gradient-free method (*i.e.*, geometric search) to move each given sample $\boldsymbol{x}$ forward the decision boundary of $f(\cdot; \boldsymbol{w})$ under the label $y$, as follows:

$$\overline{\boldsymbol{x}} = \alpha \cdot \boldsymbol{x} + (1 - \alpha) \cdot \boldsymbol{x}_y, \ s.t. \ \phi_y(\overline{\boldsymbol{x}}; \boldsymbol{w}) = 0, \tag{2}$$

where $\alpha \in [0, 1]$ is a line search parameter and $\boldsymbol{x}_y$ is a sample classified by the model $f(\cdot; \boldsymbol{w})$ as $y$.

However, the obtained boundary sample of $\boldsymbol{x}$ would be varied according to different $\boldsymbol{x}_y$. As such, we use the *closest boundary sample* of $\boldsymbol{x}$ to study the characteristics of watermarked models.

Following the previous work [30], we define the closest boundary sample of $\boldsymbol{x}$ (*i.e.*, $\overline{\boldsymbol{x}}^*$) as:

$$\overline{\boldsymbol{x}}^* \triangleq \arg\min_{\overline{\boldsymbol{x}}} ||\overline{\boldsymbol{x}} - \boldsymbol{x}||_p \ s.t. \ \phi_y(\overline{\boldsymbol{x}}; \boldsymbol{w}) = 0, \tag{3}$$

where $|| \cdot ||_{1 \leq p \leq \infty}$ is the $\ell_p$ norm.

Specifically, we can exploit the fast adaptive boundary attack (FAB) [31] to calculate the closest boundary sample. In particular, we adapt FAB to implement an iterative algorithm with gradient ascend using $\nabla_{\boldsymbol{x}} \phi_y(\boldsymbol{x}; \boldsymbol{w})$, whose update in $(t + 1)$-th iteration is as follows:

$$\overline{\boldsymbol{x}}_{t+1} = \alpha_t \cdot \boldsymbol{x}_0 + (1 - \alpha_t) \cdot \left\{ \overline{\boldsymbol{x}}_t + \beta_t \cdot \frac{\nabla_{\boldsymbol{x}} \phi_y(\overline{\boldsymbol{x}}_t; \boldsymbol{w})}{||\nabla_{\boldsymbol{x}} \phi_y(\overline{\boldsymbol{x}}_t; \boldsymbol{w})||} \right\}, \tag{4}$$

where $\beta_t$ is a positive step size, $\boldsymbol{x}_0$ is an initial point such that $\phi_y(\boldsymbol{x}_0; \boldsymbol{w}) \leq 0$ and $\alpha_t \in [0, 1]$ is chosen to ensure $\overline{\boldsymbol{x}}_{t+1}$ lies in the decision boundary as Eq. (2). In practice, $\boldsymbol{x_0}$ is randomly selected in the validation set whose label is different from $y$.

In general, using the closest boundary samples generated via Eq. (4) is mostly because they are closely related to the intrinsic property of watermarked DNNs, as shown in the next subsection.

### 3.3 The Characteristic of Boundary Gradient of Watermarked DNNs

In this section, we will show that the gradient of closest boundary samples $\nabla_{\boldsymbol{x}} \phi_y(\overline{\boldsymbol{x}}^*; \boldsymbol{w})$ (dubbed 'boundary gradients') of watermarked DNNs is closely related to the watermark patterns. Before we present our technical details, we first define $\cos\angle(\boldsymbol{x}, \nabla_{\boldsymbol{x}} f(\boldsymbol{x}_t))$ as follows:

$$\cos\angle(\boldsymbol{x}, \nabla_{\boldsymbol{x}} f(\boldsymbol{x}_t)) \triangleq \frac{< \boldsymbol{x}, \nabla_{\boldsymbol{x}} f(\boldsymbol{x}_t) >}{||\boldsymbol{x}||_2 \cdot ||\nabla_{\boldsymbol{x}} f(\boldsymbol{x}_t)||_2}. \tag{5}$$

Following previous works [32, 9], we use a model $f(\cdot; \boldsymbol{w})$ watermarked through the standard Bad-Nets backdoor attack (*i.e.*, $G_x(\boldsymbol{x}) = (\boldsymbol{1} - \boldsymbol{m}) \odot \boldsymbol{x} + \boldsymbol{m} \odot \boldsymbol{\Delta}$) [29] as a basic example to shed light on the intriguing characteristic of watermarked DNNs.

**Theorem 1** (Property of Boundary Gradient on the Closest Boundary Sample). *Assume that $\phi_y(\overline{\boldsymbol{x}}_t; \boldsymbol{w})$ is twice differentiable with a Lipschitz gradient, if $|\mathcal{D}_m| \to \infty$ and by updating $\overline{\boldsymbol{x}}_t$ in Eq. (4) with step size $\beta_t = ||\overline{\boldsymbol{x}}_t - \boldsymbol{x}_0||_2 \cdot t^{q-1}$, there exists a constant $c \geq 0$ such that*

$$\lim_{|\mathcal{D}_m| \to \infty} 1 - \cos\angle(\boldsymbol{m} \odot \boldsymbol{\delta}, \boldsymbol{m} \odot \nabla_{\boldsymbol{x}} \phi_{y_t}(\overline{\boldsymbol{x}}^*, \boldsymbol{w})) \ \leq c \cdot (t^*)^{q-1}, \tag{6}$$

*where $q \in (\frac{1}{2}, 1)$, $y_t$ is the target label (i.e., $y_t = C(G_x(\boldsymbol{x}))$), $\boldsymbol{\delta}$ is the watermark pattern (i.e., $\boldsymbol{\delta} \triangleq G_x(\boldsymbol{x}_0) - \boldsymbol{x}_0$), and $t^*$ is the number of convergence iterations of $\overline{\boldsymbol{x}}^*$'s update.*

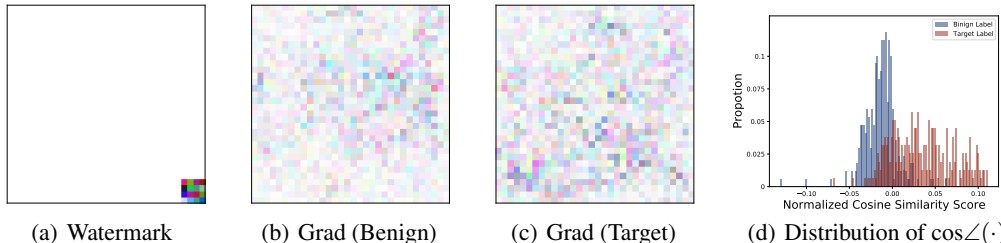

| (a) Watermark | (b) Grad (Benign) | (c) Grad (Target) | (d) Distribution of $\cos\angle(\cdot)$ |

Figure 2: **(a)** shows the watermark pattern for BadNets [29] used in our empirical study. **(b)** and **(c)** are examples of boundary gradients calculated under benign and target labels. **(d)** shows the distribution for the cosine similarity calculated over boundary gradients for benign and target labels. More empirical studies on other types of watermarks are included in the appendix.

In general, since the right side of inequality (6) decreases with the increase of $t^*$, Theorem 1 indicates that the cosine similarity between the watermark pattern $\boldsymbol{\delta}$ and the gradient calculated on $\overline{\boldsymbol{x}}_t$ located in the watermark region increases along with its update process. Its proof is in the appendix.

In the following parts, we empirically verify that the distribution of the cosine similarity between watermark patterns and boundary gradients of the closest boundary samples is different (*i.e.*, has larger values) from that on their benign versions to further justify our Theorem 1.

**Settings.** We hereby exploit BadNets-based dataset watermark [29, 6] with ResNet-18 [33] on the CIFAR-10 [4] dataset as an example for the discussion. Specifically, we watermark a sufficient amount of samples to achieve a high watermark success rate (*i.e.*, $\geq 99\%$). The watermark pattern is a $4 \times 4$ square filled with random pixels, as shown in Figure 2(a). We randomly select 400 benign samples and use Eq. (4) to generate their closest boundary version for the target label $y_t$ (*i.e.*, class '0'). To reduce the randomness caused by the selection of watermark patterns, we introduce $\overline{\cos}$ as the reference for normalization, as follows:

$$\overline{\cos} \triangleq \frac{1}{N} \sum_{i=1}^{N} \cos\angle(\boldsymbol{m} \odot \boldsymbol{\delta}_i, \boldsymbol{m} \odot \nabla_{\boldsymbol{x}} \phi_{y_t}(\overline{\boldsymbol{x}}^*, \boldsymbol{w})), \qquad (7)$$

and we calculate the normalized cosine similarity score as:

$$\widehat{\cos}\angle(\boldsymbol{m} \odot \boldsymbol{\delta}, \boldsymbol{m} \odot \nabla_{\boldsymbol{x}} \phi_{y_t}(\overline{\boldsymbol{x}}^*, \boldsymbol{w})) \triangleq \cos\angle(\boldsymbol{m} \odot \boldsymbol{\delta}, \boldsymbol{m} \odot \nabla_{\boldsymbol{x}} \phi_{y_t}(\overline{\boldsymbol{x}}^*, \boldsymbol{w})) - \overline{\cos}, \qquad (8)$$

where $\boldsymbol{\delta}_i$ is $i$-th random watermark pattern that is different from the original one (*i.e.*, $\boldsymbol{\delta}$). We generate the gradient $\nabla_{\boldsymbol{x}} \phi_y(\overline{\boldsymbol{x}}^*, \boldsymbol{w})$ of the watermarked model on the target label (dubbed 'Grad (Target)') and benign labels (dubbed 'Grad (Benign)') others than the target one, as shown in Figure 2(b) and Figure 2(c), respectively. We then calculate their normalized cosine similarity scores with 400 samples. Since the values within the boundary gradient are sparse and not evenly distributed, we follow previous work [34] to select the largest 10 values within the $\boldsymbol{m} \odot \boldsymbol{\delta}$ and $\boldsymbol{m} \odot \nabla_{\boldsymbol{x}} \phi_{y_t}(\overline{\boldsymbol{x}}^*, \boldsymbol{w}))$ to calculate their corresponding cosine similarity score. More settings details are in the appendix.

**Results.** As shown in Figure 2(d), the normalized cosine similarity scores of the target label have significantly larger values compared with those of benign labels. However, their similarity scores still have some overlap (nearly $74\%$ of the target label). It suggests that not all gradients calculated on the closest boundary samples can reflect the watermark pattern $\boldsymbol{\delta}$. It is mostly caused by the deviations introduced by the gradient estimation process under the black-box setting (as in Eq. (11)).

Nevertheless, we can still distinguish between watermarked and benign models based on their similarity distributions by comparing their *maximum* instead of random values. Specifically, suppose we define a threshold $\tau > 0$ as the maximum cosine similarity value for benign labels, and there exists $\mathbb{P}[\widehat{\cos}(\cdot) > \tau] \approx 0.26$ for the watermark model. If we randomly sample $m$ samples to calculate the (closest) boundary gradients and their corresponding normalized cosine similarity, we have:

$$\mathbb{P}(\max\{\widehat{\cos}(\cdot)_1, \widehat{\cos}(\cdot)_2, \cdots, \widehat{\cos}(\cdot)_m\} \leq \tau) = (\mathbb{P}(\widehat{\cos}(\cdot) \leq \tau))^m. \qquad (9)$$

There will be at least one sample having $\widehat{\cos}(\cdot) > \tau$ with a probability of $1 - \mathbb{P}[\widehat{\cos}(\cdot) < \tau]^m$. As such, if we sample sufficient samples (*e.g.*, 100), we will have a large chance ($\geq 99\%$) to find at least one boundary gradient larger than $\tau$ to successfully identify the watermark models. These phenomena inspire the design of our ZeroMark method, as proposed in the next section.

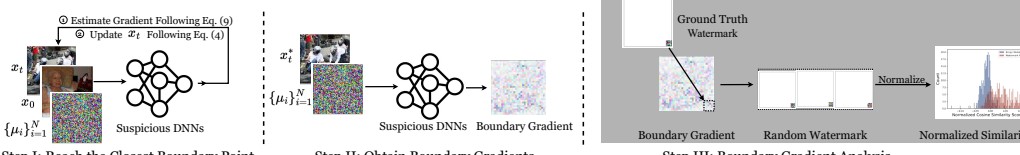

Figure 3: The main pipeline of our ZeroMark, which consists of three main steps. In the first steps, ZeroMark applies Eq. (4) and Eq. (10) to generate the closest boundary samples. In the second step, ZeroMark estimates the boundary gradients for generated boundary samples. In the third step, ZeroMark first leverages the ground truth watermark pattern and random watermark patterns to calculate the normalized cosine similarity following Eq. (7) and Eq. (8). After that, ZeroMark uses the distribution of similarity scores to conduct $t$-test for dataset ownership verification. In particular, in the third stage, patterns (*i.e.*, random and ground truth watermarks) displayed in the gray background are only available to the defender and inaccessible to the suspicious model.

## 4 Methodology

In this section, we describe the threat model and the technical details of our ZeroMark method.

**Threat Model.** Consistent with previous DOV methods [7, 6, 8], we assume that dataset owners will watermark the original dataset to generate its watermarked version. The dataset owner has full knowledge of watermark patterns and the validation samples used for ownership verification. In the verification stage, given a suspicious model, they will examine whether it was trained on the protected dataset under the label-only black-box setting, where they can only query the model with verification samples via model API and get its predicted labels, without accessing its intermediate results (*e.g.*, gradients) and model parameters.

**The Main Pipeline of ZeroMark.** In general, our ZeroMark consists of three main steps, as shown in Figure 3. ZeroMark first generates the (closest) boundary samples of the suspicious model. Then, it calculates the boundary gradients of the generated boundary samples. ZeroMark conducts dataset ownership verification via boundary gradient analysis in the third step.

**Step 1. Generate Closest Boundary Samples.** ZeroMark follows Eq. (4) to optimize the closest boundary samples $\overline{x}^*$ iteratively with gradient decent. In particular, we exploit Monte Carlo method to estimate $\nabla_{\boldsymbol{x}}\phi_y(\overline{x}_t; \boldsymbol{w})$ to address the challenge in our considered label-only black-box scenarios, where the gradients for given inputs are inaccessible. The overall process is as follows:

$$\overline{\boldsymbol{x}}_{t+1} = \alpha_t \cdot \boldsymbol{x}_0 + (1 - \alpha_t) \cdot \left\{ \overline{\boldsymbol{x}}_t + \beta_t \cdot \frac{\frac{1}{N}\sum_{i=1}^{N}\phi_y(\overline{\boldsymbol{x}}_t + \kappa \cdot \mu_i; \boldsymbol{w}) \cdot \mu_i}{||\frac{1}{N}\sum_{i=1}^{N}\phi_y(\overline{\boldsymbol{x}}_t + \kappa \cdot \mu_i; \boldsymbol{w}) \cdot \mu_i||} \right\}, \quad (10)$$

where $\beta_t$ is the step size, $\boldsymbol{x}_0$ is an initial point such that $\phi_y(\boldsymbol{x}_0; \boldsymbol{w}) \leq 0$, $\alpha_t \in [0, 1]$ is chosen to ensure $\overline{\boldsymbol{x}}_{t+1}$ lies in the decision boundary as Eq. (2), $\{\mu_i\}_{i=1}^{N} \sim N(0, 1)$ are $N$ random noises $i.i.d$ sampled from the standard Gaussian distribution, and $\kappa$ is a fixed positive parameter (*i.e.*, 0.01).

**Step 2. Calculate Boundary Gradients.** Once the closest boundary sample $\overline{\boldsymbol{x}}^*$ is generated, we can also exploit Monte Carlo method to estimate its gradient (dubbed 'boundary gradient'), as follows:

$$\nabla_{\boldsymbol{x}}\phi_y(\overline{\boldsymbol{x}}^*; \boldsymbol{w}) \approx \frac{1}{N}\sum_{i=1}^{N}\phi_y(\overline{\boldsymbol{x}}^* + \kappa \cdot \mu_i; \boldsymbol{w}) \cdot \mu_i. \quad (11)$$

**Step 3. Boundary Gradient Analysis.** After obtaining the boundary gradients, ZeroMark first calculates the cosine similarity based on the available watermark pattern $\boldsymbol{\delta}$ and obtains boundary gradients. To further mitigate the variance caused by the watermark patterns, we create several random watermark patterns and follow Eq. (8) to normalize the calculated cosine similarity. After that, motivated by the characteristic described in Section 3.3, where the cosine similarity of the boundary gradients on the target label $y_t$ has a significantly larger value compared with that of benign labels, we design a hypothesis-test-guided method to conduct ownership verification based on the range of cosine similarity for verification, as follows.

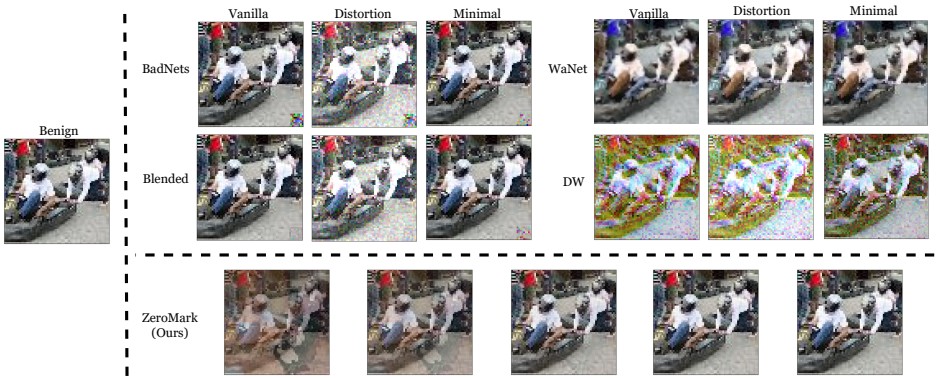

Figure 4: The example of verification samples across different watermarks (*i.e.*, BadNets, Blended, WaNet, DW) and verification methods (*i.e.*, Vanilla, Minimal, Distortion) on Tiny-ImageNet.

**Proposition 1.** *Suppose $\widehat{cos}\angle(\boldsymbol{m}\odot\boldsymbol{\delta},\boldsymbol{m}\odot\nabla_{\boldsymbol{x}}\phi_{y_t}(\overline{\boldsymbol{x}}^*,\boldsymbol{w}))$ is the posterior normalized cosine similarity between boundary gradients and the available trigger pattern of the suspicious model $f(\cdot,\boldsymbol{w})$. Let variables $P_b$ and $P_d$ denote the largest $Q\%$ over $\{\widehat{cos}\angle(\boldsymbol{m}\odot\boldsymbol{\delta},\boldsymbol{m}\odot\nabla_x\phi_{y_t}(\overline{\boldsymbol{x}}_i^*,\boldsymbol{w}))\}_{i=1}^m$ and $\{\widehat{cos}\angle(\boldsymbol{m}\odot\boldsymbol{\delta},\boldsymbol{m}\odot\nabla_x\phi_{y\neq y_t}(\overline{\boldsymbol{x}}_i^*,\boldsymbol{w}))\}_{i=1}^m$, respectively. Given the null hypothesis $H_0 : P_b = P_d + \tau$ ($H_1 : P_b > P_d + \tau$) where the hyper-parameter $\tau \in [0,1]$, we claim that the suspicious model is trained on the protected dataset (with $\tau$-certainty) if and only if $H_0$ is rejected.*

In general, we randomly select $m$ (*i.e.*, 500) validation samples evenly distributed across different classes to generate boundary gradients for the watermarked and benign labels. Then, we conduct the pairwise $t$-test [35] and calculate its p-value. The null hypothesis $H_0$ is rejected if the p-value is smaller than the significance level $\alpha$. Besides, we also calculate the *confidence score* $\Delta P = P_b - P_d$ to represent the verification confidence. *The larger the $\Delta P$, the more confident the verification.*

## 5 Experiments

In this section, we conduct experiments on CIFAR-10 [4] and Tiny-ImageNet [36] datasets with ResNet18 and ResNet-34 [33], respectively. More results with other settings are in our appendix.

### 5.1 The Performance of Verification Samples Generated by ZeroMark

**Settings.** We hereby compare our ZeroMark method to two straightforward baseline methods, including **(1)** verification with minimal watermark (dubbed 'Minimal') and **(2)** verification with distorted watermark (dubbed 'Distortion'). These methods intend to protect the information of dataset-specified watermarks by perturbing the original watermarks. We also provide the results of verification with benign samples (dubbed 'Benign') and verification with original dataset-specified watermarked samples (dubbed 'Vanilla') for reference. We evaluate each verification method on four dataset watermark techniques, including two patch-based watermarks [29, 37] and two input-specific ones [38, 9]. Regarding the implementation of existing watermark techniques, we follow their default settings. The example of verification samples across different watermarks and verification methods is shown in Figure 4. Please find more details in the appendix.

**Evaluation Metrics.** We adopt mean square error (MSE), neuron activation similarity (NAS), and mutual information (MI) to measure the degree to which the dataset-specified watermarks are disclosed during the verification stage. Specifically, the MSE is defined as the mean square error between verification and their corresponding watermarked samples in the region of watermark patterns. NAS is calculated as the cosine similarity in the neuron activation map between verification and their corresponding watermarked samples. MI is calculated based on the distribution of verification and their corresponding watermarked samples. More details are in our appendix.

**Results.** As shown in Table 1, our approach produces larger cosine similarity scores of the target label than benign labels. We show the results regarding the watermark-disclosed degree for different approaches in Table 2-3. The results show that our approach can reach the most minor watermark-

Table 1: The averaged largest $Q\%$ cosine similarity of our method on different watermarks.

| Dataset→ | CIFAR-10 | | | | | TinyImageNet | | | | |
|---|---|---|---|---|---|---|---|---|---|---|
| Label↓, Watermark→ | BadNets | Blended | WaNet | DW | Ave. | BadNets | Blended | WaNet | DW | Ave. |
| Benign | 0.028 | 0.030 | 0.022 | 0.028 | 0.027 | 0.026 | 0.021 | 0.029 | 0.027 | 0.026 |
| Target | 0.102 | 0.368 | 0.131 | 0.099 | 0.174 | 0.148 | 0.334 | 0.131 | 0.123 | 0.194 |

Table 2: The performance on CIFAR-10. In particular, we mark the best results in bold while the value within the underline denotes the second-best results (except the benign samples).

| Metric→ | MSE ($\uparrow$) | | | | NAS ($\downarrow$) | | | | MI ($\downarrow$) | | | |
|---|---|---|---|---|---|---|---|---|---|---|---|---|
| Watermark→ 
 Method↓ | BadNets | Blended | WaNet | DW | BadNets | Blended | WaNet | DW | BadNets | Blended | WaNet | DW |
| Benign | 0.394 | 0.197 | 0.077 | 0.309 | 0.597 | 0.617 | 0.609 | 0.665 | 15.9 | 19.7 | 22.3 | 28.5 |
| Vanilla | 0 | 0 | 0 | 0 | 0.830 | 0.801 | 0.767 | 0.824 | 64.3 | 56.1 | 58.2 | 61.7 |
| Minimal Distortion | 0.193 | 0.171 | 0.121 | 0.197 | 0.797 | 0.769 | 0.721 | 0.743 | 54.8 | 51.5 | 44.0 | 52.3 |
|  | 0.286 | **0.251** | 0.087 | **0.301** | 0.770 | 0.774 | 0.701 | 0.769 | 56.6 | 53.8 | 35.4 | 44.5 |
| ZeroMark (Ours) | **0.392** | 0.202 | **0.199** | 0.246 | **0.646** | **0.671** | **0.688** | **0.689** | **18.1** | **24.4** | **28.6** | **29.7** |

Table 3: The performance on Tiny-ImageNet. In particular, we mark the best results in bold while the value within the underline denotes the second-best results (except the benign samples).

| Metric→ | MSE ($\uparrow$) | | | | NAS ($\downarrow$) | | | | MI ($\downarrow$) | | | |
|---|---|---|---|---|---|---|---|---|---|---|---|---|
| Watermark→ 
 Method↓ | BadNets | Blended | WaNet | DW | BadNets | Blended | WaNet | DW | BadNets | Blended | WaNet | DW |
| Benign | 0.396 | 0.189 | 0.076 | 0.298 | 0.497 | 0.561 | 0.589 | 0.629 | 27.6 | 29.8 | 28.2 | 31.4 |
| Vanilla | 0 | 0 | 0 | 0 | 0.817 | 0.808 | 0.763 | 0.804 | 68.1 | 59.0 | 64.7 | 67.4 |
| Minimal Distortion | 0.187 | 0.096 | 0.077 | 0.189 | 0.768 | 0.782 | **0.696** | 0.749 | 57.3 | 54.6 | 49.9 | 54.7 |
|  | 0.263 | **0.227** | 0.079 | **0.227** | 0.773 | 0.745 | 0.738 | 0.773 | 61.4 | 55.7 | 39.2 | 48.3 |
| ZeroMark (Ours) | **0.314** | 0.204 | **0.171** | 0.201 | **0.662** | **0.697** | 0.704 | **0.698** | **27.7** | **33.2** | **30.5** | **31.7** |

Table 4: The verification performance of our method on different watermarks.

| Dataset→ | | CIFAR-10 | | | Tiny-ImageNet | | |
|---|---|---|---|---|---|---|---|
| Watermark↓ | Metric↓, Scenario→ | Independent-W | Independent-M | Malicious | Independent-W | Independent-M | Malicious |
| BadNets | $\Delta P$ | 0.012 | 0.013 | 0.081 | 0.011 | 0.012 | 0.127 |
|  | p-value | 1.00 | 1.00 | $10^{-45}$ | 1.00 | 1.00 | $10^{-58}$ |
| Blended | $\Delta P$ | 0.010 | 0.013 | 0.35 | 0.016 | 0.012 | 0.313 |
|  | p-value | 1.00 | 1.00 | $10^{-67}$ | 1.00 | 1.00 | $10^{-64}$ |
| WaNet | $\Delta P$ | 0.028 | 0.012 | 0.102 | 0.022 | 0.011 | 0.110 |
|  | p-value | 0.80 | 1.00 | $10^{-53}$ | 0.90 | 1.00 | $10^{-55}$ |
| DW | $\Delta P$ | 0.023 | 0.014 | 0.071 | 0.030 | 0.002 | 0.101 |
|  | p-value | 0.88 | 1.00 | $10^{-12}$ | 0.74 | 1.00 | $10^{-49}$ |

disclosed degree in almost all cases. The improvement is significant compared to the vanilla DOV methods. These results verify the effectiveness and security of our ZeroMark.

## 5.2 The Performance of Dataset Ownership Verification via ZeroMark

**Settings.** We evaluate our ZeroMark for ownership verification under three scenarios, including **(1)** independent watermark (dubbed 'Independent-W'), **(2)** independent model (dubbed 'Independent-M'), and **(3)** unauthorized dataset training (dubbed 'Malicious'). In the first case, we used ZeroMark to verify the suspicious model affected with other watermark patterns; In the second case, we test the benign model with our ZeroMark; In the last case, we use ZeroMark to verify the watermarked model with corresponding ground truth watermark samples. Notice that only the last case should be regarded as having unauthorized dataset use. More setting detail are described in the appendix.

**Evaluation Metrics.** Following the settings in [7, 6], we use $\Delta P \in [-1, 1]$ and p-value $\in [0, 1]$ for the evaluation. For the first two independent cases, a small $\Delta P$ and a large $p$-value are expected. In contrast, for the third one, the larger $\Delta P$ and the smaller the p-value, the better the verification.

**Results.** As shown in Table 4, our method can achieve accurate dataset ownership verification in all cases. Specifically, our approach can identify the unauthorized dataset usage with high confidence (*i.e.*, p-value $\ll 0.01$ for 'Malicious' case), while not misjudging when there is no unauthorized dataset utilization (*i.e.*, p-value $\gg 0.1$ for 'Independent-W' and 'Independent-M').

## 5.3 Ablation Study

We hereby study the effects of two key hyper-parameters. More results are in our appendix.

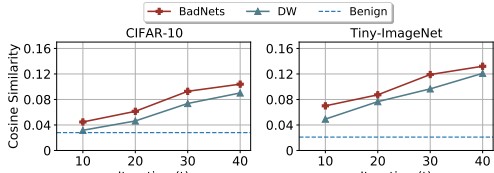

Figure 5: Effects of iteration size $t$.

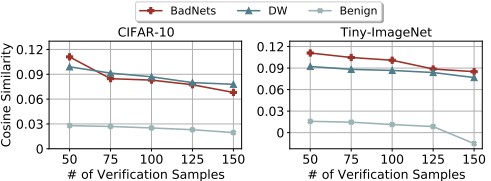

Figure 6: Effects of verification sample size.

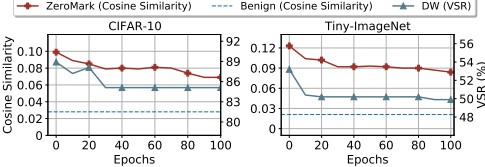

Figure 7: Robustness against fine-tuning.

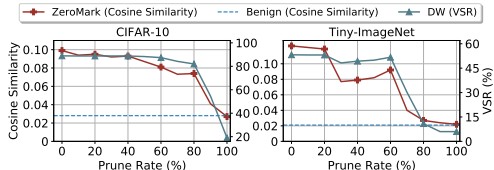

Figure 8: Robustness against model pruning.

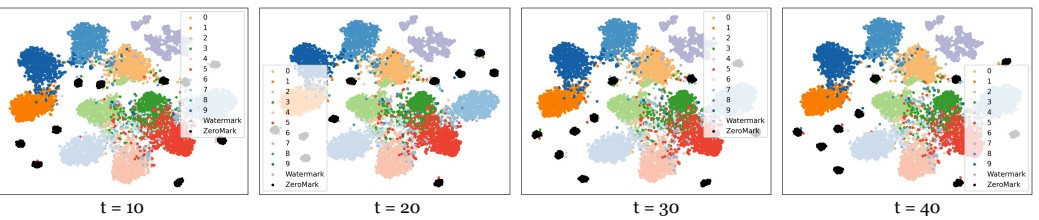

Figure 9: The t-SNE results with varied optimization iterations $t$ for the embedding features of benign, watermark, and ZeroMark samples extracted from watermarked DNNs.

**Effects of Optimization Iterations** $t$. We exploit BadNets and DW as the representatives of patch-based and sample-specific watermarks to study the effects of $t$ on both CIFAR-10 and Tiny-ImageNet datasets. As shown in Figure 5, the cosine similarity scores increase with the increase of $t$. With $t \geq 30$, we can easily distinguish watermarked and benign models.

**Effects of the Largest Number of Verification Samples** (*i.e.*, $m \times Q\%$). As shown in Figure 6, with the number of verification samples used in the $t$-test increases, the cosine similarity decreases while the benign models' cosine similarity remains stable on both datasets. However, our method can still have a promising cosine similarity even with many verification samples (*e.g.*, 150).

### 5.4 The Resistance to Potential Adaptive Methods

We hereby select domain watermark to evaluate the robustness of ZeroMark, as the domain watermark itself is sufficiently robust against this defense. Following the previous work [7], we here evaluate the robustness of our ZeroMark under fine-tuning [39] and model pruning [40]. As shown in Figure 7, fine-tuning has minor effects on our method. In Figure 8, we can see ZeroMark performs resilient against model pruning as its efficacy decreases along with the domain watermark. The results of our resistance to other methods are in our appendix.

### 5.5 A Closer Look to the Effectiveness of our ZeroMark

In this section, we intend to further explore the mechanisms behind the effectiveness of our Zero-Mark. Specifically, we adopt t-SNE [41] to visualize the feature distribution of watermark samples, benign samples, and samples generated by ZeroMark of watermarked DNNs. As shown in Figure 9, with varied optimization iterations $t$ for generating (closest) boundary samples, samples generated by ZeroMark can always stay far away from the watermark samples' distribution, which demonstrates ZeroMark can prevent disclosing the watermark information from the watermark samples.

## 6 Potential Limitations and Future Directions

ZeroMark can conduct dataset ownership verification without disclosing the watermark pattern. However, as the first work towards a secure verification process of dataset ownership verification (DOV) methods, we have to admit that we still have some potential limitations.

Firstly, we must admit that our method requires additional time to conduct ownership verification since it needs to generate some boundary samples and their gradients under the black-box setting. For example, it takes nearly 30 mins for verification on CIFAR-10. While this cost is acceptable in practice to a large extent, we will discuss how to accelerate our ZeroMark in our future work.

Secondly, ZeroMark currently can only perform effectively for watermark techniques with a pre-defined target label ($e.g.$, BadNets [29], etc. For other watermark techniques, which have no pre-defined target label ($i.e.$, UBW [7]), ZeroMark can not conduct boundary gradient analysis and, therefore, is not a feasible solution. Moreover, through extensive experimental evaluation, Zero-Mark is shown to perform more effectively for Blended watermark, compared with other watermark techniques. Such observations inspire us to improve the effectiveness of ZeroMark by designing the potential watermark patterns in a blended manner. Finally, the boundary gradient analysis step in ZeroMark may incur variance among different labels' samples. In practice, we can release such variance by training a surrogate model with the protected dataset and calculating cosine similarity between the corresponding boundary gradients and the target watermark pattern with samples from different labels. Then we use the calculated cosine similarity using samples from different labels under the surrogate model to adjust the cosine similarity calculated under the suspicious model. Specifically, we can adjust the cosine similarity calculated with samples from a specific label $t$ by subtracting it from the average cosine similarity calculated with samples from the same label $t$ under the surrogate model. We will further discuss these issues in our future work.

Thirdly, we can only empirically verify that malicious dataset users cannot recover dataset-specified watermarks based on our boundary samples. We will try to prove it theoretically in the future.

Fourthly, we are currently focusing on convolutional neural networks ($e.g.$, ResNet and VGG) and the continuous image modality. In general, the success of our approach on other model structures depends on two factors: **(1)** whether the studied dataset watermarking method ($e.g.$, BadNets) can successfully watermark these models and **(2)** whether we can conduct effective 'adversarial attacks' to find boundary samples on these models. Based on existing work related to backdoor attacks/dataset watermarking [42, 9] and adversarial attacks [43, 44], these factors are all met. As such, our method can fundamentally generalize to other models ($e.g.$, transformer) as well. As for the generalizability to other (discrete) data formats like tabular or text, the main challenge lies in how to design effective adversarial attacks to them for finding the closest boundary samples (as in Eq.(10)). In particular, there are already some relevant works [45, 46] confirming its feasibility. Accordingly, our ZeroMark can be naturally adapted to other discrete data formats ($e.g.$, text and scientific data [47, 48, 49, 50, 51]). We will discuss them in our future work.

## 7 Conclusion

In this paper, we revisited existing DOV methods and revealed that their underlying assumption regarding the verification phase does not necessarily hold in practice. Accordingly, directly using dataset-specified watermarks for verification is insecure. Motivated by these findings, we proposed *ZeroMark* to conduct ownership verification without disclosing them. Our method was inspired by our empirical and theoretical findings of the intrinsic property of DNNs trained on the watermarked dataset. We conducted experiments on benchmark datasets, verifying the effectiveness of our Zero-Mark and its resistance to potential adaptive methods. We hope our work can provide a new angle of dataset ownership verification to facilitate more secure and trustworthy dataset sharing.

## Acknowledgment

This work was partially supported by NSF IIS 2347592, 2347604, 2348159, 2348169, DBI 2405416, CCF 2348306, and CNS 2347617.

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

**Appendix**

## Table of Contents

# A Proof for Theorem 1

We follow the previous work [32] to use a model $f(\cdot; \boldsymbol{w})$ infected with the basic backdoor-based watermark (i.e., $G_x = (1 - m) \odot \Delta + m \odot \boldsymbol{x}$) [29] as a basic example to shed light on the intriguing characteristic of watermark model.

**Theorem 1** (Property of Boundary Gradient on the Closest Boundary Sample). *Assume that $\phi_y(\overline{\boldsymbol{x}}_t; \boldsymbol{w})$ is twice differentiable with a Lipschitz gradient, if $|\mathcal{D}_m| \to \infty$ and by updating $\overline{\boldsymbol{x}}_t$ in Eq. (4) with step size $\beta_t = ||\overline{\boldsymbol{x}}_t - \boldsymbol{x}_0||_2 \cdot t^{q-1}$, there exists a constant $c \geq 0$ such that*

$$\lim_{|\mathcal{D}_m| \to \infty} 1 - cos\angle(\boldsymbol{\delta}, \nabla_{\boldsymbol{x}}\phi_{y_t}(\overline{\boldsymbol{x}}^*, \boldsymbol{w})) \leq c \cdot (t^*)^{q-1} \tag{12}$$

*where $q \in (\frac{1}{2}, 1)$, $y_t$ is the target label (i.e., $y_t = C(G_x(\boldsymbol{x}))$), $\boldsymbol{\delta}$ is the watermark pattern (i.e., $\boldsymbol{\delta} \triangleq G_x(\boldsymbol{x}_0) - \boldsymbol{x}_0$), and $t^*$ is the number of convergence iterations of $\overline{\boldsymbol{x}}^*$'s update.*

*Proof.*

Recall in Eq. (4), we update $\overline{\boldsymbol{x}}_t$ for each $t - 1$ th iteration as:

$$\overline{\boldsymbol{x}}_{t+1} = \alpha_t \cdot \boldsymbol{x}_0 + (1 - \alpha_t) \cdot \left\{ \overline{\boldsymbol{x}}_t + \beta_t \frac{\nabla_{\boldsymbol{x}}\phi_y(\overline{\boldsymbol{x}}_t; \boldsymbol{w})}{||\nabla_{\boldsymbol{x}}\phi_y(\overline{\boldsymbol{x}}_t; \boldsymbol{w})||} \right\}, \tag{13}$$

Let the step size $\beta_t$ in Eq. (4) as $t^{-q}||\boldsymbol{x_t} - \boldsymbol{x_0}||$, we have the distance ratio for updating Eq. (4) as:

$$\frac{||\overline{\boldsymbol{x}}_{t+1} - \boldsymbol{x_0}||^2}{||\overline{\boldsymbol{x}}_t - \boldsymbol{x_0}||^2} = \frac{||(1 - \alpha)(\frac{t^{-q}||\overline{\boldsymbol{x}}_t - \boldsymbol{x_0}||\nabla_{\boldsymbol{x}}\phi_y(\overline{\boldsymbol{x}}_t; \boldsymbol{w})}{||\nabla_{\boldsymbol{x}}\phi_y(\overline{\boldsymbol{x}}_t; \boldsymbol{w})||} + \overline{\boldsymbol{x}}_t - \boldsymbol{x_0})||_2^2}{||\overline{\boldsymbol{x}}_t - \boldsymbol{x_0}||_2^2} \tag{14}$$

With a second-order tylor expansion, we have:

$$\phi_y(\overline{\boldsymbol{x}}_t; \boldsymbol{w}) = \ < \nabla_{\boldsymbol{x}}\phi_y(\overline{\boldsymbol{x}}_t; \boldsymbol{w}), \overline{\boldsymbol{x}}_{t+1} - \overline{\boldsymbol{x}}_t > + \frac{1}{2}(\overline{\boldsymbol{x}}_{t+1} - \overline{\boldsymbol{x}}_t)^T H_t(\overline{\boldsymbol{x}}_{t+1} - \overline{\boldsymbol{x}}_t) = 0 \tag{15}$$

Combining these Eq. (4) and Eq. (15), we have:

$$< \nabla_{\boldsymbol{x}}\phi_y(\overline{\boldsymbol{x}}_t; \boldsymbol{w}), -\alpha v_t + \tau_t \nabla_{\boldsymbol{x}}\phi_y(\overline{\boldsymbol{x}}_t; \boldsymbol{w}) > + \tag{16}$$

$$\frac{1}{2}(-\alpha v_t + \tau_t \nabla_{\boldsymbol{x}}\phi_y(\overline{\boldsymbol{x}}_t; \boldsymbol{w}))^T H_t(-\alpha v_t + \tau_t \nabla_{\boldsymbol{x}}\phi_y(\overline{\boldsymbol{x}}_t; \boldsymbol{w})) = 0, \tag{17}$$

where we define $v_t$ as $\overline{\boldsymbol{x}}_t - \boldsymbol{x_0} + \tau_t \nabla_{\boldsymbol{x}}\phi_y(\overline{\boldsymbol{x}}_t; \boldsymbol{w})$ and $\tau_t$ as $t^{-q}\frac{||\overline{\boldsymbol{x}}_t - \boldsymbol{x_0}||}{||\nabla_{\boldsymbol{x}}\phi_y(\overline{\boldsymbol{x}}_t; \boldsymbol{w})||}$.

Solving for $\alpha$, we have:

$$\alpha \geq \frac{\nabla_{\boldsymbol{x}}\phi_y(\overline{\boldsymbol{x}}_t; \boldsymbol{w})^T(\tau_t^2 H_t + 2\tau_t I)\nabla_{\boldsymbol{x}}\phi_y(\overline{\boldsymbol{x}}_t; \boldsymbol{w})}{2\nabla_{\boldsymbol{x}}\phi_y(\overline{\boldsymbol{x}}_t; \boldsymbol{w})^T(I + \tau_t H_t)v_t}. \tag{18}$$

Therefore, we can get:

$$(1 - \alpha)^2 \leq \left( \frac{r_t + \frac{3}{2}t^{-q}L\frac{||d_t||_2}{||\nabla_{\boldsymbol{x}}\phi_y(\overline{\boldsymbol{x}}_t; \boldsymbol{w})||_2}}{r_t + t^{-q}(1 + \frac{3}{2}L\frac{||d_t||_2}{||\nabla_{\boldsymbol{x}}\phi_y(\overline{\boldsymbol{x}}_t; \boldsymbol{w})||_2})} \right), \tag{19}$$

where $d_t = \overline{\boldsymbol{x}}_t - \boldsymbol{x_0}$ and $r_t := cos\angle(\overline{\boldsymbol{x}}_t - \boldsymbol{x}_0, \nabla_{\boldsymbol{x}}\phi_y(\overline{\boldsymbol{x}}_t; \boldsymbol{w})) = \frac{<\overline{\boldsymbol{x}}_t - \boldsymbol{x_0}, \nabla_{\boldsymbol{x}}\phi_y(\overline{\boldsymbol{x}}_t; \boldsymbol{w})>}{||\overline{\boldsymbol{x}}_t - \boldsymbol{x_0}||_2||\nabla_{\boldsymbol{x}}\phi_y(\overline{\boldsymbol{x}}_t; \boldsymbol{w})||_2} = \frac{<d_t, \nabla_{\boldsymbol{x}}\phi_y(\overline{\boldsymbol{x}}_t; \boldsymbol{w})>}{||d_t||_2||\nabla_{\boldsymbol{x}}\phi_y(\overline{\boldsymbol{x}}_t; \boldsymbol{w})||_2}$.

Let $k_t := \frac{3}{2}L\frac{||d_t||_2}{||\nabla_t||_2}$. Then $k_t$ can be bounded when $||\nabla_t||_2 \geq C$ and $q \geq \frac{1}{2}$, thus we have:

$$\frac{||x_{t+1} - x^*||_2^2}{||x_t - x^*||_2^2} \leq \left( \frac{r_t + \beta_t k_t}{r_t + \beta_t(1 + k_t)} \right)^2 \cdot (\beta_t^2 + 2\beta_t r_t + 1) \tag{20}$$

Motivated by previous work [52], solve Eq. (20) and have :

$$\sum_{t=1}^{\infty} c_1 t^{-q} \frac{1 - r_t^2}{r_t} - c_2 t^{-2q} \leq \infty, \tag{21}$$

where $c_1, c_2$ are two positive constants, thus the above equation is $o(t - 1)$.

When $q \in (\frac{1}{2}, 1)$, we have:

$$\frac{1 - r_t^2}{r_t} = o(t^{q-1}). \tag{22}$$

Therefore, we have:

$$1 - cos\angle(d_t, \nabla_{\boldsymbol{x}}\phi_y(\overline{\boldsymbol{x}}_t; \boldsymbol{w})) \leq c \cdot t^{q-1}, \tag{23}$$

Notably, Eq. (4) with step size as $t^{-q}||\overline{\boldsymbol{x}}_t - \boldsymbol{x_0}||$ converges a stationary point of Eq. (3). Motivated by proof for **Lemma 3** in [34], when $\overline{\boldsymbol{x}}_t$ is optimized to a stationary point (*i.e.*, $\overline{\boldsymbol{x}}^*$) in $t^*$ and if $\overline{\boldsymbol{x}}_t$ belongs to the watermark label $y_t$, we have:

$$\lim_{|\mathcal{D}|_m \to \infty} \mathbb{E}\left[m \odot \overline{\boldsymbol{x}}^* - m \odot \boldsymbol{x_0}\right] = \mathbb{E}\left[d_t\right] = m \odot \boldsymbol{\Delta} - m \odot \boldsymbol{x_0} \tag{24}$$

$$= m \odot \boldsymbol{\delta}, \tag{25}$$

and

$$\lim_{|\mathcal{D}|_m \to \infty} \frac{||(1 - m) \odot (\overline{\boldsymbol{x}}^* - \boldsymbol{x_0})||}{||\overline{\boldsymbol{x}}^* - \boldsymbol{x_0}||} = 0. \tag{26}$$

Hence, when $|\mathcal{D}|_m \to \infty$, we have:

$$\frac{< \boldsymbol{m} \odot d_t, \boldsymbol{m} \odot \nabla_{\boldsymbol{x}}\phi_y(\overline{\boldsymbol{x}}^*; \boldsymbol{w}) >}{||\boldsymbol{m} \odot d_t||||\boldsymbol{m} \odot \nabla_{\boldsymbol{x}}\phi_y(\overline{\boldsymbol{x}}^*; \boldsymbol{w})||} = \frac{< \boldsymbol{m} \odot \boldsymbol{\delta}, \boldsymbol{m} \odot \nabla_{\boldsymbol{x}}\phi_y(\overline{\boldsymbol{x}}^*; \boldsymbol{w}) >}{||\boldsymbol{m} \odot \boldsymbol{\delta}||||\boldsymbol{m} \odot \nabla_{\boldsymbol{x}}\phi_y(\overline{\boldsymbol{x}}^*; \boldsymbol{w})||}, \tag{27}$$

therefore, for the watermark label (*i.e.*, $y_t$):

$$\lim_{|\mathcal{D}_m| \to \infty} 1 - cos\angle(\boldsymbol{m} \odot \boldsymbol{\delta}, \boldsymbol{m} \odot \nabla_x\phi_{y_t}(\overline{\boldsymbol{x}}^*, \boldsymbol{w})) = 1 - cos\angle(\boldsymbol{m} \odot d_t, \boldsymbol{m} \odot \nabla_{\boldsymbol{x}}\phi_y(\overline{\boldsymbol{x}}^*; \boldsymbol{w}))$$
$$\leq c \cdot (t^*)^{q-1}. \tag{28}$$

## B  Detailed Settings for Empirical Study in Section  3.3

In the Section. 3.3, we select ResNet-18 and CIFAR-10 as the evaluated model and benchmark. We select the class '0' as the watermark class and inject watermark samples with 10% watermark ratio to ensure the verification success rate $\geq 99\%$. We randomly select 300 samples from the validation set across classes. We use these selected validation samples to generate boundary points for watermark and benign labels labels following Eq. (4). In particular, for the boundary point of watermark label $y_t$, we set $\boldsymbol{x_0}$ as samples from classes different from $y_t$ and set $\boldsymbol{x_t}$ as samples from the watermark label. As for the boundary point of benign labels, we set $\boldsymbol{x_0}$ as samples from the watermark class and set $\boldsymbol{x_t}$ as samples from the benign labels. As such , there should be 400 boundary gradients for the watermark or benign label. We then calculate the boundary gradients following the gradient estimation process as Eq. (10).

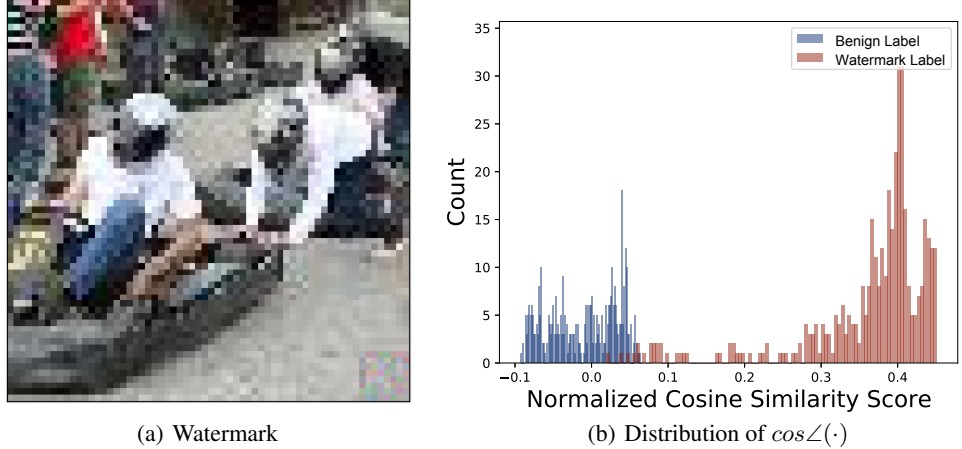

(a) Watermark

(b) Distribution of $cos\angle(\cdot)$

Figure 10: The results of using blended watermark.

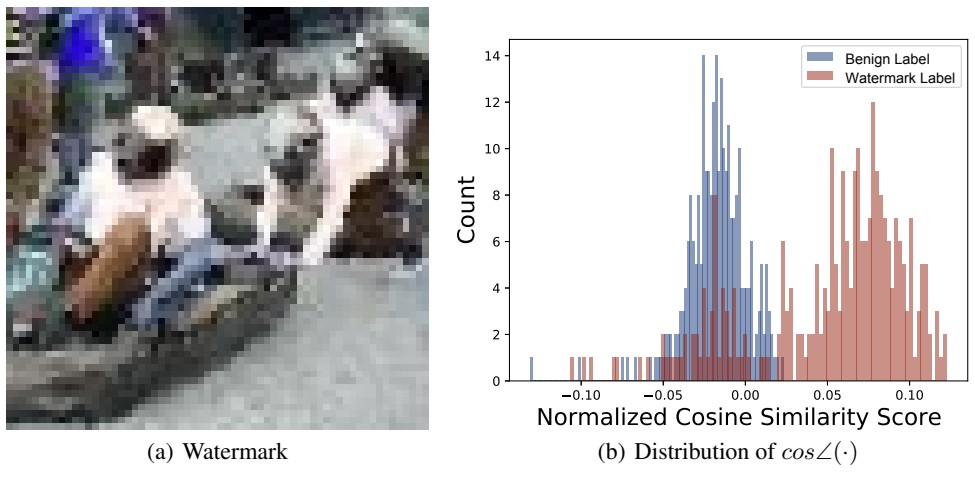

(a) Watermark

(b) Distribution of $cos\angle(\cdot)$

Figure 11: The results of using WaNet watermark.

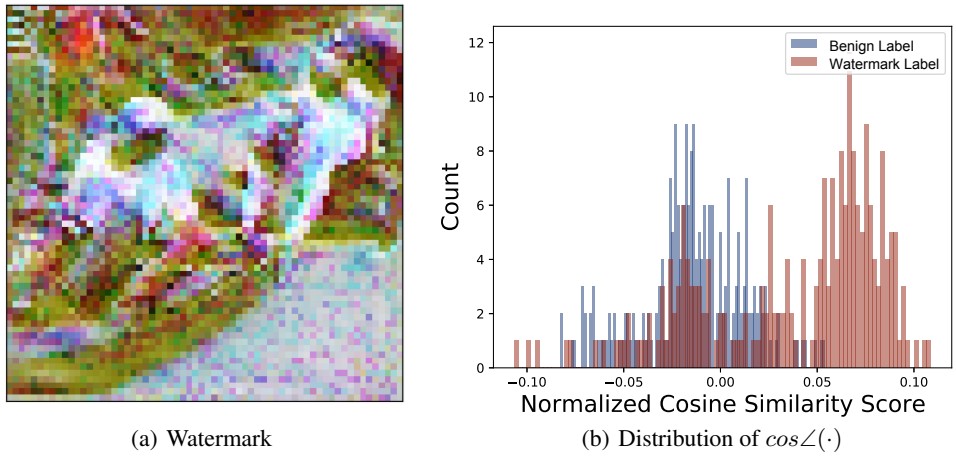

(a) Watermark

(b) Distribution of $cos\angle(\cdot)$

Figure 12: The results of using domain watermark.

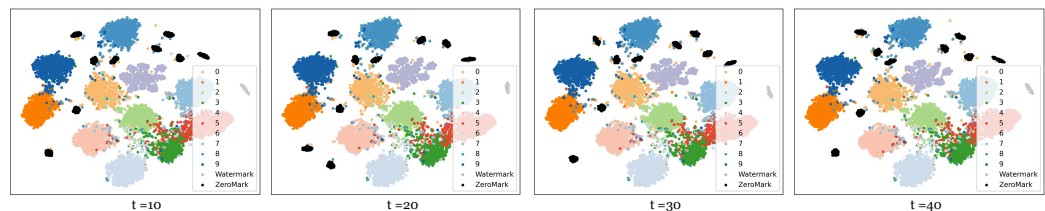

Figure 13: t-SNE clustering results for the blended watermark.

## C Additional Results for Empirical Study in Section 3.3

We also conduct same empirical studies on additional three different types of watermarks (*i.e.*, Blended [37],WaNet [38] and Domain Watermark [9]) to validate the characteristic of boundary gradients for watermarked and benign labels. Specifically, we conduct empirical studies with CIFAR-10 using ResNet-18 and select '0' as the watermark label. We can see that in all these three watermark patterns, the distributions of cosine similarity for watermarked labels have significantly large ranges compared with that of benign labels. Interestingly enough, we find that the distribution of cosine similarity for Blended watermark has a more obvious separation with that of benign labels compared with other watermark patterns. We will speculate the reason for this phenomenon in our future work.

## D The Detailed Process for Mitigating the Variance

We here describe how to mitigate the variance caused by the watermark patterns and the iterative gradient estimation in details.

**Mitigate the Variance Caused by the Watermark Patterns.**    Based on the available ground truth watermark pattern $\boldsymbol{\delta}$, we create several (*i.e.*, 6) artifact watermark patterns $\{\boldsymbol{\delta_i}\}_{i=1}^{10}$ which have the same location map (*i.e.* $\boldsymbol{m}$) as the ground truth watermark but filled with different random noise. We calculate the baseline $\overline{cos}$ as:

$$\overline{cos} := \frac{1}{N} \sum_{i=1}^{N} cos\angle(m \odot \boldsymbol{\delta_i}, m \odot \nabla_x \phi_{y_t}(\overline{\boldsymbol{x}}^*, \boldsymbol{w})), \tag{29}$$

and we calculate the normalized cosine similarity score as:

$$\widehat{cos}\angle(m \odot \boldsymbol{\delta}, m \odot \nabla_x \phi_{y_t}(\overline{\boldsymbol{x}}^*, \boldsymbol{w})) := cos\angle(m \odot \boldsymbol{\delta}, m \odot \nabla_x \phi_{y_t}(\overline{\boldsymbol{x}}^*, \boldsymbol{w})) - \overline{cos}. \tag{30}$$

**Mitigatwe the Variance in the Iterative Gradient Estimation Procedure.**    During the procedure of gradient estimation for Eq. (10), the estimated gradients could yield variance among iterations for the gradient estimation process. Therefore, to mitigate such variance, we propose to average the estimated gradients over iterations for the gradient estimation process, which can be formulated as:

$$\nabla_x \phi_{y_t}(\overline{\boldsymbol{x}}^*, \boldsymbol{w})) := \frac{1}{t} \sum_{t=0}^{t} \nabla_x \phi_{y_t}(\overline{\boldsymbol{x}}_t, \boldsymbol{w})) \tag{31}$$

## E Detailed description for Evaluation Metrics

We here describe the metrics for evaluating each approach in details. With loss of generality, we here define each give watermark sample as:

$$\boldsymbol{x}'_i = \boldsymbol{x_i} + \boldsymbol{t_i}, \tag{32}$$

where $\boldsymbol{x_i}$ and $\boldsymbol{t_i}$ represent the benign sample and the corresponding watermark pattern. We suppose $\boldsymbol{t_i}$ is located at the dims of $[j : k](j < k)$. The evaluation metrics, including mean square error (MSE), neuron activation similarity (NAS) and mutual information (MI) are defined as below:

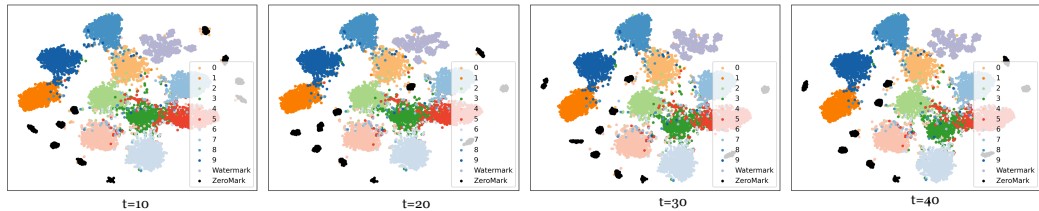

Figure 14: t-SNE clustering results for the WaNet watermark.

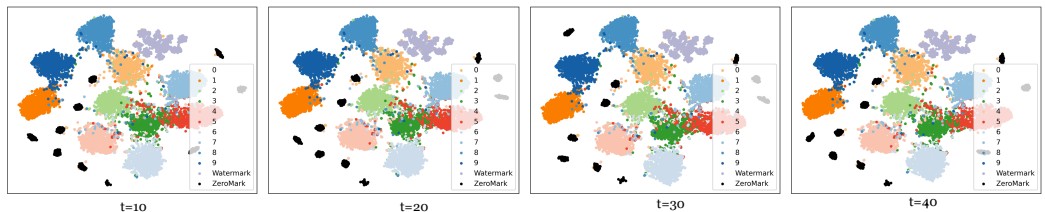

Figure 15: t-SNE clustering results for the domain watermark.

---

**Algorithm 1** The main process of our ZeroMark.

---

1: **Input:** Validation samples $\{\boldsymbol{x_i^t}, y_t\}_{i=1}^m$ from the watermark label $y_t$; Validation samples $\{\boldsymbol{x_i^o}, y_i^o\}_{i=1}^m$ from the labels other than the watermark label; The suspicious model $f(\cdot; \boldsymbol{w})$
2: **for** l=1,2,... **do**
3:     Generate (closest) boundary points and boundary gradients for the watermark label following Eq. (4): Set $\{\boldsymbol{x_i^t}, y_t\}_{i=1}^m$ as $\boldsymbol{x_y}$ and set $\{\boldsymbol{x_i^o}, y_i^o\}_{i=1}^m$ as $\boldsymbol{x_0}$.
4:     Generate (closest) boundary points and boundary gradients for the benign label following Eq. (4): Set $\{\boldsymbol{x_i^t}, y_t\}_{i=1}^m$ as $\boldsymbol{x_0}$ and set $\{\boldsymbol{x_i^o}, y_i^o\}_{i=1}^m$ as $\boldsymbol{x_y}$.
5: **end for**
6: Calculate the corresponding cosine similarities for watermark and benign labels following Eq. (8).
7: Select largest $m*Q\%$ cosine similarities for watermark and benign labels for T-test following Proposition 1.

---

- MSE: $= \frac{1}{N} \sum_{i=1}^N \sqrt{(\boldsymbol{x_i'}[j:k] - \boldsymbol{x_v}[j:k])^2}$.

- NAS: $= \frac{1}{N} \sum_{i=1}^N \cos\angle(F(\boldsymbol{x_i'}), F(x_v)))$.

- MI:$= \mathbb{E}_{p(\boldsymbol{z_v}, \boldsymbol{z'})}[\log p(\boldsymbol{z'}|\boldsymbol{z_v})] - \mathbb{E}_{p(\boldsymbol{z_v})p(\boldsymbol{z'})}[\log p(\boldsymbol{z'}|\boldsymbol{z_v})]$.

where $\boldsymbol{x_v}$, $F(\cdot)$ represent the verification samples, feature extractor for the corresponding watermark model. $\boldsymbol{z}$ represents the feature extracted by the watermark model. Since ZeroMark uses several perturbed boundary points for calculating the boundary gradient for each given sample, we average their values for computing each metric. Notably, we follow previous work [53] to estimate MI by calculating its upper bound, as follows:

$$I(\boldsymbol{z}; \hat{\boldsymbol{z}}) = \mathbb{E}_{p(\boldsymbol{z}, \hat{\boldsymbol{z}})}\left[\log \frac{p(\hat{\boldsymbol{z}}|\boldsymbol{z})}{p(\hat{\boldsymbol{z}})}\right] \leq \mathbb{E}_{p(\boldsymbol{z}, \hat{\boldsymbol{z}})}[\log p(\hat{\boldsymbol{z}}|\boldsymbol{z})] - \mathbb{E}_{p(\boldsymbol{z})p(\hat{\boldsymbol{z}})}[\log p(\hat{\boldsymbol{z}}|\boldsymbol{z})]. \quad (33)$$

# F   The Detailed Algorithm for ZeroMark

We put the detailed algorithm for ZeroMark as follows:

# G  Experiments Details

## G.1  Datasets

We evaluate our approach on two widely-adopted benchmark datasets (i.e., CIFAR-10 [4], Tiny-ImgaeNet [36]). We here describe each benchmark dataset in detail.

**CIFAR-10.**  CIFAR-10 dataset contains 10 labels, 50,000 training samples, and 10,000 validation samples. The training and validation samples are distributed evenly across each label. Each sample is resized as $32 \times 32$ by default.

**Tiny-ImageNet.**  Tiny-ImageNet dataset contains 200 labels, 100,000 training samples, and 10,000 validation samples. The training and validation samples are distributed evenly across each label. Each sample is resized as $64 \times 64$ by default.

**Evaluated Watermarks.**  In our experiments, we evaluate four types of watermark, including BadNets [29], Blended [37], WaNet [38], and domain watermark [9]. The visual demonstration for each watermark is shown in Sec. 4. For BadNets, we implement a $4 \times 4$ and $8 \times 8$ triggers for CIFAR-10 and Tiny-ImageNet. The trigger is filled with random noise. For Blended watermark, we implement a $4 \times 4$ and $8 \times 8$ triggers for CIFAR-10 and Tiny-ImageNet. We set the transparency ratio as 0.2 throughout experiments. As for WaNet, we use `BackdoorBox`[2] [54] to build the watermarked model with its default configurations. For Domain Watermark, we implement it following its released code[3]. We set the watermark rate $\gamma$ as 0.1 consistent with previous work [7] for training different watermark models.

## G.2  Training Configurations

To train DNN models, we use Adam optimizer [55] with the initial learning rate as 0.01. The watermark models evaluated in our experiments can achieve $\geq 92.26\%$ and $56.8\%$ accuracy on validation dataset for CIFAR-10 and Tiny-ImageNet tasks. We use six NVIDIA RTX 2080 Ti GPUs for performing experiments.

# H  Detailed Configurations for Comparison Approaches

We here describe the comparison approaches in details, as follows.

**Existing DOV.**  We follow DOV approaches [29, 37, 38, 9] to directly exploit the watermark samples for verification.

**Minimal Watermark.**  Inspired by previous work [56] for Trojan detection, which applies reverse engineering to generate the pseudo trigger patterns with minimize size while preserving their attack efficacy. Specifically, we generate the minimal watermark for the watermark sample $\boldsymbol{x'} = \boldsymbol{x} + \boldsymbol{\delta}$ following:

$$\min_{\boldsymbol{m_\delta}} \ell(f(\boldsymbol{m_\delta} \odot \boldsymbol{\delta} + (1 - \boldsymbol{m_\delta}) \odot \boldsymbol{x}; \boldsymbol{w}), y_t) + ||\boldsymbol{m_\delta}||_2, \tag{34}$$

we follow Neural Cleanse [56]'s configurations for conducting optimization on Eq. (34). For input-specific watermark patterns (*e.g.*, WaNet [38], DW [9]), we conduct Eq. (34) on each watermark sample. As such verification samples with the minimal watermark can be formulated as:

$$x' = \boldsymbol{m_\delta} \odot \boldsymbol{\delta} + (1 - \boldsymbol{m_\delta}) \odot \boldsymbol{x}. \tag{35}$$

---

[2]`https://github.com/THUYimingLi/BackdoorBox.git`
[3]`https://proceedings.neurips.cc/paper_files/paper/2023/hash/aa6287ca31ae1474ea802342d0c8ba63-Abstract-Conference.html`

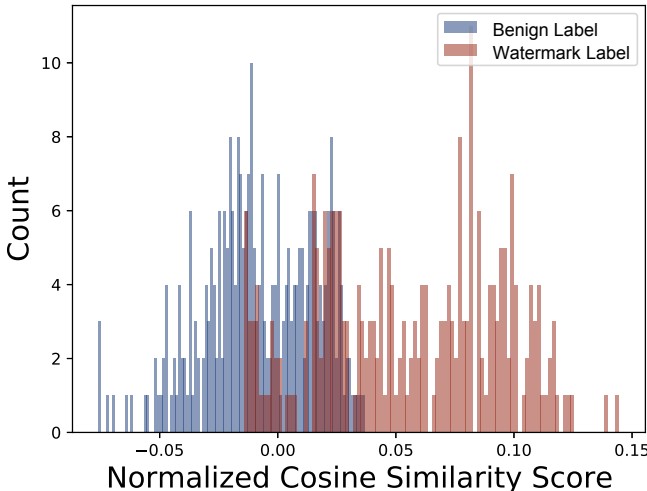

Figure 16: The distribution of consine similarity of boundary gradients for watermark and benign labels under VGG-19.

**Watermark with Distortion.** Motivated by previous work [32], which reveals that watermark samples can perform resilient against certain magnitude of random noise distortion, we thus propose to add maximum magnitudes of random noise for watermark samples to hinder the watermark pattern while preserving the watermark's efficacy. We generate the noisy watermark samples via:

$$\boldsymbol{x'} = Proj(\boldsymbol{x'} + a * \boldsymbol{\mu}), \tag{36}$$

where $\boldsymbol{\mu} \sim N(0,1)$ is Gaussian noise and $Proj$ is the projection function to constrain $\boldsymbol{x'} + a * \boldsymbol{\mu}$ into $[0,1]$. We solve $a$ using grid search to find the largest $a$ while preserving the verification success rate for watermark samples.

## I  Additional Results for Experiments

We here show additional results in our experiments. We perform t-SNE clustering analysis for other three types of watermark (*i.e.*, Blended, WaNet, Domain Watermark) with varied optimization iterations $t$. The results are shown in Figure 13, Figure 14 and Figure 15. The additional visual demonstrations for boundary samples within the verification procedure are shown in Figure 17, Figure 18, Figure 19 and Figure 20. We also evaluate ZeroMark with different model architectures. Specifically, we here evaluate ZeroMark using VGG-19 [57] with CIFAR-10 task and the configurations are consistent with Section. 3.3. The results are shown in Figure 16, which demonstrates that ZeroMark can still perform effective on VGG-19 models.

## J  The Resistance to More Adaptive Attacks

We here investigate whether ZeroMark can perform robustness against the potential adaptive attack for recovering or unlearning the watermark pattern.

**Recovering the Watermark Pattern from the Boundary Samples.** We here explore the potential adaptive attack for recovering the watermark pattern. Since we leverage gradient estimation via aggregating the random noise for conducting boundary gradient analysis, we here consider an adaptive attack by aggregating corresponding boundary samples for each input sample $\boldsymbol{x_0}$ to recover the watermark pattern. We conduct experiments using ResNet-18 under CIFAR-10 task and we evaluate the adaptive attack with Domain Watermark [9]. The results are shown in Figure 21 and Figure 22.

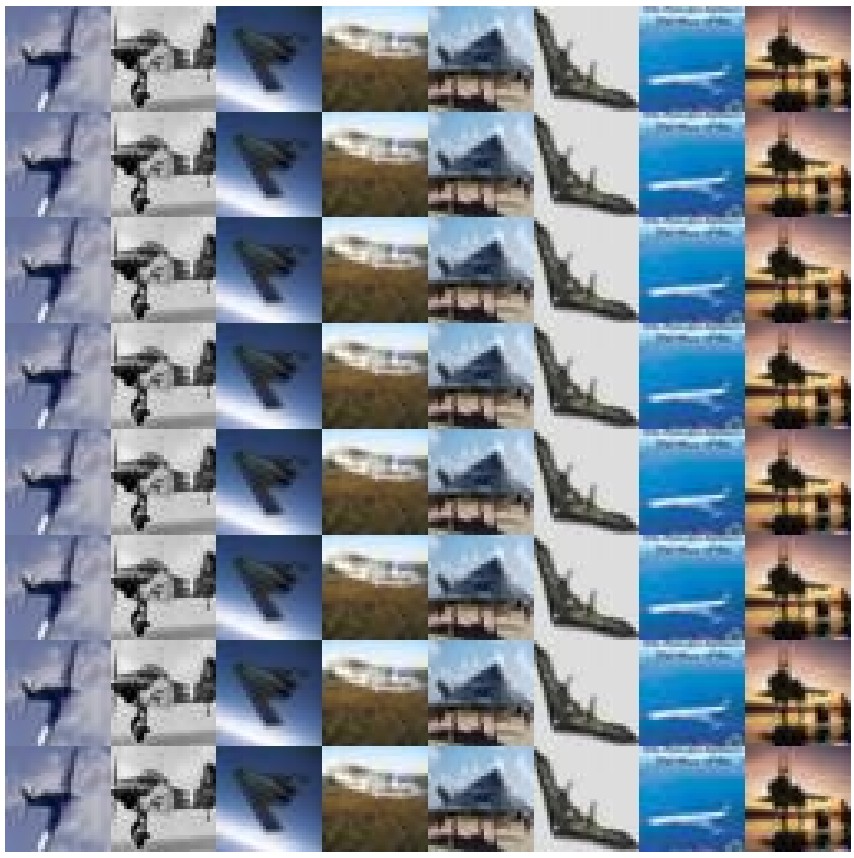

Figure 17: Visual demonstration of ZeroMark for BadNets watermark.

We find that the aggregated boundary samples can not reveal the watermark pattern from both visual and clustering analysis.

**Unlearn the Watermark through Boundary Samples.** We here consider whether we can follow [56] to unlearn the watermark pattern via boundary samples. We retrain the suspicious model with 500 boundary samples and label them as their original label (*i.e.*, label for $x_0$) along with the training data. We find that the accuracy of the suspicious model drops from 92.3% to 90.2% and the verification success rate for domain watermark drops from 88.6% to 75.1%, and the ZeroMark can still achieve the averaged largest Q% as 0.076 for the target label, which significantly larger than that of the benign labels(*i.e.*,0.028). This results demonstrate that ZeroMark can prevent disclosing watermark patterns during the verification procedure within DOV.

## K    Reproducibility Statement

In the appendix, we provide detailed descriptions of the datasets, models, training and evaluation settings, and computational facilities. We provide the codes and model checkpoints for reproducing the main experiments of our evaluation in the supplementary material.

## L    Societal Impacts

In this paper, we focus on the copyright protection of public datasets. Specifically, we reveal that the verification process of existing DOV methods is not secured and propose using boundary samples to conduct verification without disclosing the watermark. This work has no general ethical issues since our method is purely defensive and does not reveal any new vulnerabilities of DNNs.

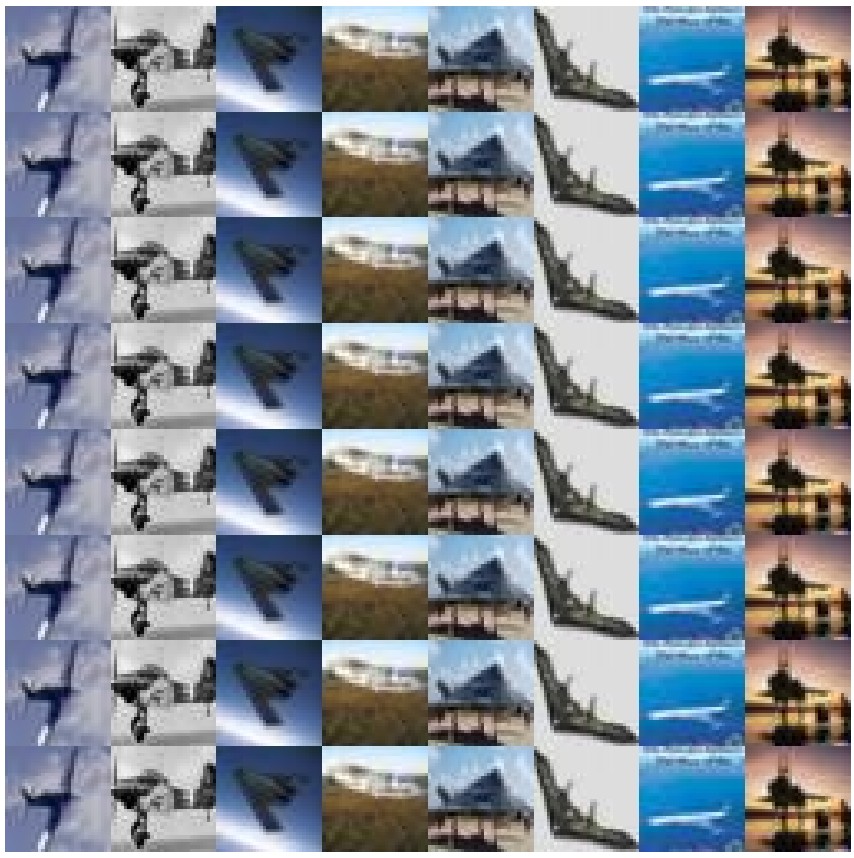

Figure 18: Visual demonstration of ZeroMark for Blended watermark.

## M    Discussions about Adopted Data

In this paper, all adopted samples are from the open-sourced datasets (*i.e.*, CIFAR-10, Tiny-ImageNet). The Tiny-ImageNet dataset may contain a few human-related images. We admit that we modified a few samples for watermarking and verification. However, our research treats all samples the same and the verification samples and modified samples have no offensive content. Accordingly, our work fulfills the requirements of these datasets and has no privacy violation.

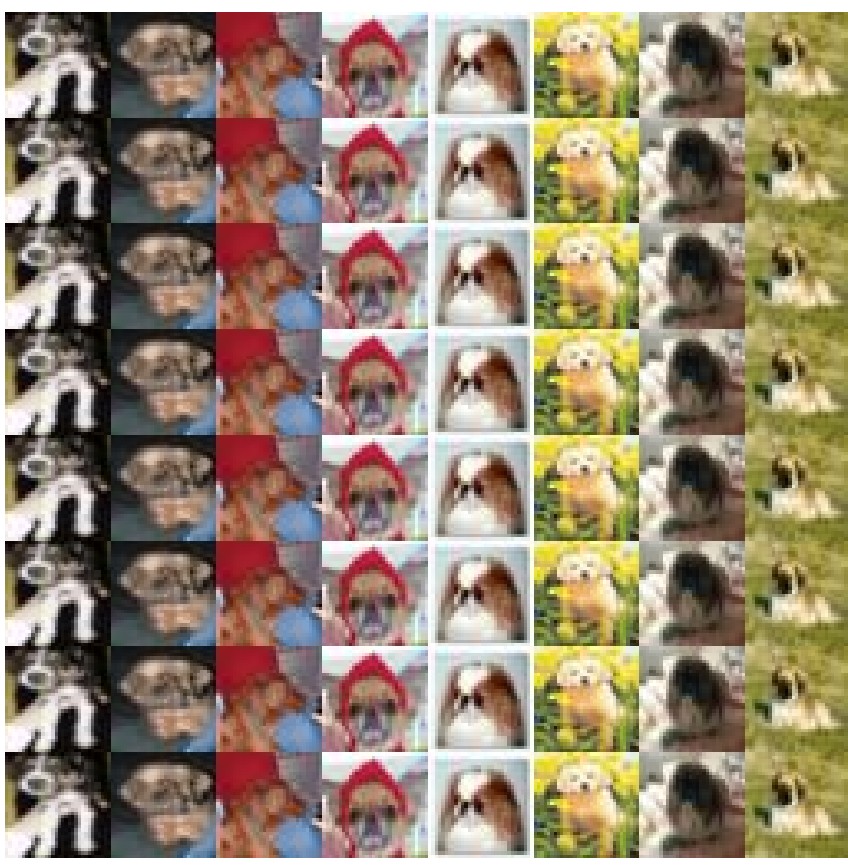

Figure 19: Visual demonstration of ZeroMark for WaNet watermark.

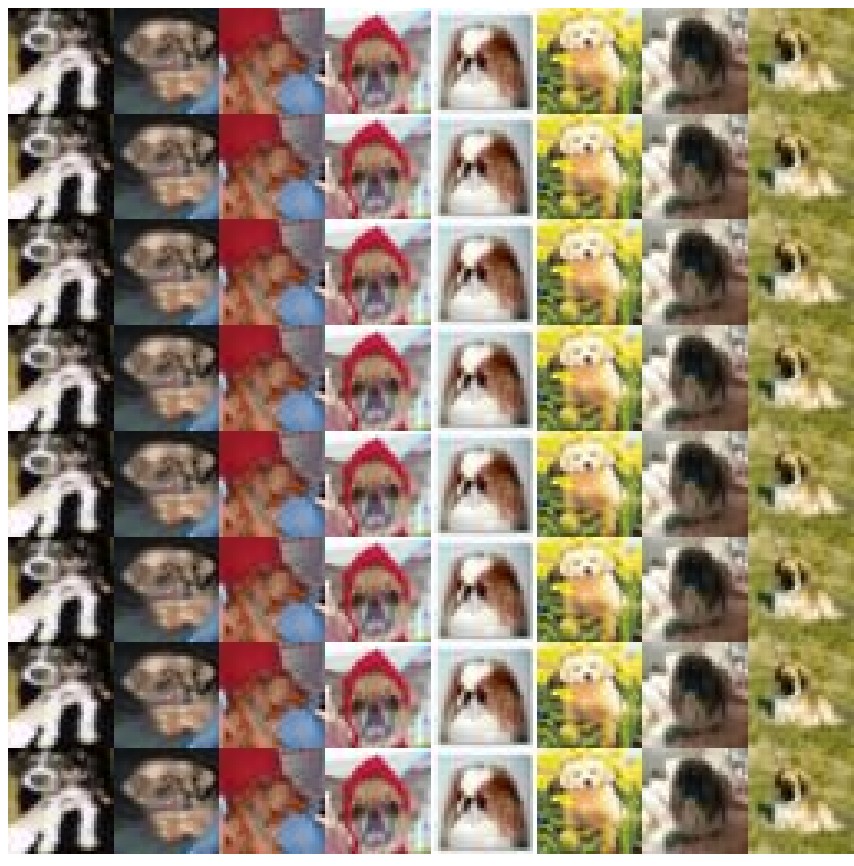

Figure 20: Visual demonstration of ZeroMark for domain watermark.

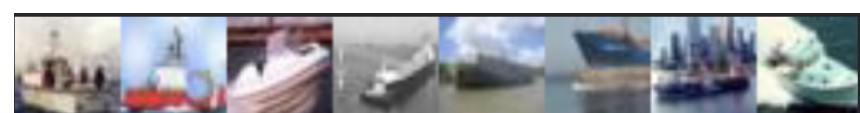

Figure 21: The visual demonstration of the aggregated boundary samples.

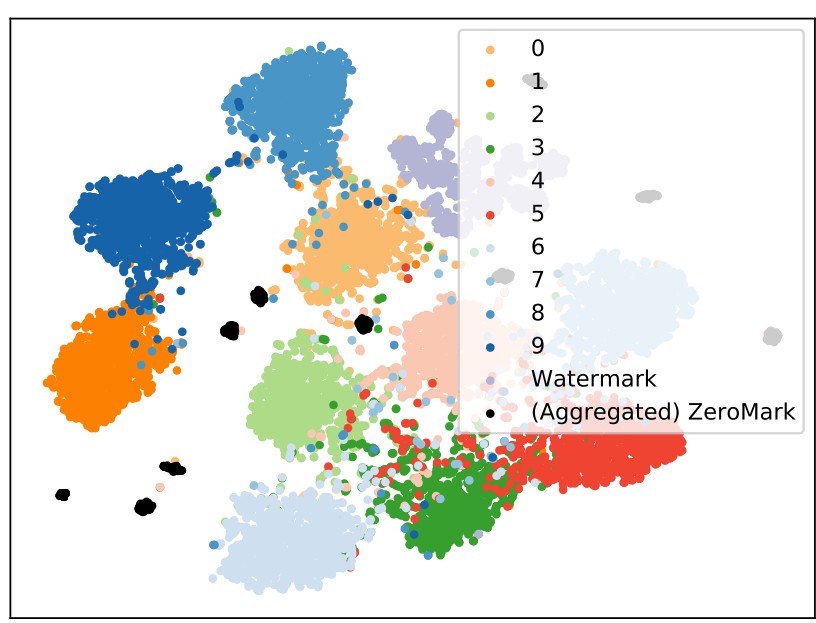

Figure 22: The t-SNE clustering results for aggregated boundary samples.

