# OpenReview forum: "ZeroMark: Towards Dataset Ownership Verification without Disclosing Watermark"
_NeurIPS.cc/2024/Conference — NeurIPS 2024 poster_

### Official Review · Reviewer_s8Mh · 2024-07-08

**Soundness:** 3
**Presentation:** 3
**Contribution:** 2
**Rating:** 3
**Confidence:** 4

**Summary:**

This paper proposes a new dataset ownership verification method, ZeroMark, which calculates the boundary gradient between the benign and reconstructed images to verify the authorship. Extensive experiments are conducted on two datasets with four backdoor attack methods to evaluate the ZeroMark.

**Strengths:**

1) This paper proposes a new dataset ownership verification (DOV) method that provides a new angle to protect the copyright of our valuable datasets.

2) The authors revisit existing DOV methods in detail that increase the readability notably.

**Weaknesses:**

1) In Lines 40~44, the authors mention that the problem contained in existing methods is the leakage of the watermark pattern. This issue is not reasonable since keeping the safety of the watermark is a significant requirement for all watermarking methods. The embedded watermark can also be removed when ZeroMark releases the original watermark pattern.

2) As shown in Fig.2 (d), there is no clear bond between the benign samples and target labels. Distinguishing the distribution to clarify the authorship is may not believable among various images.

3) The novelty of this paper is limited. Using backdoor behaviors [1] or their distributions to protect the dataset shows similar forms.

4) In Eq. (5), is it correct to calculate the cosine similarity between the image x and its gradient?

5) In Fig. (4), the trigger shown in Blended is wrong. Blended injects a Hello Kitty or a pair of sunglasses as the trigger.

6) Why does the smaller MSE denote the better performance? From my view, the MSE indicates the difference between the original watermark and the boundary region. The small value denotes that the boundary region is more like the watermark.

7) Although the authors discussed some limitations in future work, the issues of robustness and efficiency of ZeroMark also hinder the practicability. ZeroMark is a backdoor-based DOV method, the state-of-the-art backdoor defense methods should be included in the experimental section.

[1] Guo J, Li Y, Wang L, et al. Domain watermark: Effective and harmless dataset copyright protection is closed at hand[J]. Advances in Neural Information Processing Systems, 2024, 36.

**Questions:**

In line 27, the authors believe that DOV is the only feasible way to protect the copyright of public datasets currently. However, Unlearnable examples have been studied to prevent unauthorized data exploitation [1]. How do you think about it?

[1] Huang H, Ma X, Erfani S M, et al. Unlearnable Examples: Making Personal Data Unexploitable[C]//International Conference on Learning Representations (2021).

**Limitations:**

The authors have discussed the limitations.

---

> ### Author Rebuttal · Authors · 2024-08-06
>
> Dear Reviewer s8Mh, thank you very much for your careful review of our paper and thoughtful comments. We are encouraged by your positive comments on our **good soundness and presentation**, **extensive experiments**, and **novelty**. We hope the following responses can help clarify potential misunderstandings and alleviate your concerns.
>
>
> ---
> # The Clarification of Our Main Focus
>
> In this paper, we propose a new DOV method **by introducing a new verification process** without disclosing dataset-specified watermarks **instead of proposing new dataset watermarks** as in previous works. Our method can be incorporated to enhance existing dataset watermarks to support a privacy-preserving verification process.
>
> ---
> **Q1**: The embedded watermark can also be removed when ZeroMark releases the original watermark pattern.
>
> **R1**: We are deeply sorry to cause your misunderstanding that we want to clarify.
>
> - As mentioned in the previous clarification, **our ZeroMark is a new verification method instead of a new watermarking method**. As such, **our work is different from all existing DOV works**, whose main focus is proposing new watermarks.
> - **ZeroMark does not release the original watermark pattern**. Our ZeroMark queries the suspicious model with benign images and their boundary versions, instead of directly using watermarked samples.
> -  **ZeroMark prevents information leakage for watermark patterns and resists to watermark reconstruction and unlearning attacks**.
>     - As shown in Tabel 2-3, ZeroMark prevents information leakage of watermark patterns for existing watermark techniques since the verification contains limited information about trigger patterns. As such, it is difficult for adversaries to reconstruct dataset-specified trigger patterns based on boundary samples.
>     - As shown in Section 5.4 and Appendix J, ZeroMark resists to unlearning with boundary samples.
>
> ---
>
> **Q2**: As shown in Fig.2 (d), there is no clear bond between the benign samples and target labels.
>
> **R2**: Thank you for this insightful question! **We distinguished between watermarked and benign models** based on their similarity distributions **by comparing their maximum instead of random values**. Specifically, **we selected the largest Q% samples within each distribution**. Please kindly find more detailed explanations in Line180-188.
>
>
> ---
>
> **Q3**: The novelty of this paper is limited. Using backdoor behaviors [1] or their distributions to protect the dataset shows similar forms.
>
> **R3**: Thank you for these comments and we do understand your concern.
>
> - As we mentioned in the clarification at the beginning, **we focus on the verification process instead of watermarking process** that is discussed in all previous works.
> - **We explore the non-disclosure requirement of the verification process for the first time**.
> - We empirically and theoretically discover an intrinsic property of watermarked DNNs regarding boundary samples.
>
> ---
>
> **Q4**: In Eq. (5), is it correct to calculate the cosine similarity between the image x and its gradient?
>
> **R4**: We are deeply sorry that we may lead you to some misunderstandings that we want to clarify here. **Eq.(5) is used to illustrate the definition of cosine similarity**. We never implements Eq.(5) in our method.
>
> ---
> **Q5**: Blended use Hello Kitty or sunglasses as the trigger.
>
> **R5**: Thanks your comments!
> - Arguably, **the main contribution of Blended is the blended strategy for generating stealthy poisoned images instead of the specific trigger patterns** (e.g., Hello Kitty and sunglasses). Our settings simply follow those used in existing works.
> - **Random noise pattern can validate the generalizability of ZeroMark**. It can validate that ZeroMark is effective and generalizable, not specific to a certain watermark pattern.
> - To further alleviate your concerns, we evaluate it with Hello Kitty. As shown in Table 5-6 in the uploaded PDF, **our method is still highly effective under this setting**.
>
> ---
> **Q6**: Why does the smaller MSE denote the better performance?
>
> **R6**: Larger MSE indicates the better performance.
>
> ---
>
> **Q7**: ZeroMark is a backdoor-based DOV method, the state-of-the-art backdoor defense methods should be included.
>
> **R7**: Thank you for this insightful comment!
>
> - **Our method can also be incorporated in non-backdoor-based dataset watermarks** (e.g., domain watermark).
> - Even when used in backdoor-based watermarks, **our method naturally escapes most backdoor detection methods** because our query samples (i.e., boundary samples) are benign and do not contain trigger patterns.
> - We have evaluted ZeroMark against two classical backdoor defenses (i.e., fine-tuning and pruning) in Section 5.3.
> - To further alleviate your concerns, we conduct experiments on SCALE-UP, STRIP and ShrinkPad using CIFAR-10 with ResNet-18. **Our method resists to them** with the AUROC (0.52, 0.50, 0.52).
>
> ---
> **Q8**: Unlearnable examples can also be used to prevent unauthorized data exploitation.
>
> **R8**: Thanks for bringing this paper to us!
> - **We admit that unlearnable examples can also be used to protect data, although in a different aspect** (availability v.s. tracebility). We will modify this sentence to avoid potential misunderstandings or over-claims.
> - To further alleviate your concern, we **summarize the differences between unlearnable examples and dataset ownership verification**, as follows.
>     - Unlearnable examples aim to make DNNs fail to have a high accuracy trained on the protected data. **It is usually used to protect data published on social media and cannot be used to detect unauthorized dataset users**.
>     - **Unlearnable samples usually need to watermark the whole dataset or at least the majority parts of the victim dataset**, whereas DOV methods only need to modify a few samples.
>     - **Unlearnable samples can only be used to prevent unauthorized dataset usage**, whereas DOV methods can also trace/attribute unauthorized users.
> ---

---

> > ### Comment · Reviewer_s8Mh · 2024-08-09
> > **Response to rebuttal**
> >
> > I thank the authors for clarifying some of the questions. As there is no reaction regarding my concerns and suggestions in the weakness section. I am sticking with my original rating.
> >
> > 1. ZeroMark essentially injects a BadNet (or Blended) trigger into the dataset to achieve dataset authorship verification, while only the inputted watermarked samples have been changed to boundary samples during the verification phase. An important issue that should be considered is that these triggers have very low stealthiness. The effectiveness of this method would be significantly degraded if malicious users use Sota defense methods when training their models, which has already been demonstrated by current backdoor defense methods [1, 2].
> >
> > 2. For the unclear boundary, an important and in-depth ablation study about the Q% has been lost.
> >
> > 3. Keeping the correct explanation about existing works (e.g., Blended) and equations (e.g., Eq. (5)) is crucial. If you "never" implement Eq. (5) in your experiment, why do you list it in your paper?
> >
> > 4. Domain Watermark is a backdoor-based DOV method that employs clean-label backdoor attacks to implement dataset protection.
> >
> > 5. Please don't over-claim your research topic. Unlearnable examples are a more feasible way to prevent unauthorized dataset usage than ZeorMark. The ZeroMark can only check whether someone has trained their models on your released dataset. If you check it, unauthorized dataset usage has already occurred.
> >
> >
> > I am thrilled to hear various opinions from the authors and other reviewers.
> >
> > [1] Gao K, Bai Y, Gu J, et al. Backdoor defense via adaptively splitting poisoned dataset[C]//Proceedings of the IEEE/CVF Conference on Computer Vision and Pattern Recognition. 2023: 4005-4014.
> >
> > [2] Zhu M, Wei S, Zha H, et al. Neural polarizer: A lightweight and effective backdoor defense via purifying poisoned features[J]. Advances in Neural Information Processing Systems, 2024, 36.

---

> > > ### Author Response · Authors · 2024-08-09
> > > **Responses to Post-rebuttal Comments (Round I) [Part 1]**
> > >
> > > Dear Reviewer s8Mh,
> > >
> > > We sincerely thank you for your timely follow-up feedback! We hereby provide more explanations to clarify some potential misunderstandings and further alleviate your remaining concerns.
> > >
> > > ---
> > > **Q1**: ZeroMark essentially injects a BadNet (or Blended) trigger into the dataset to achieve dataset authorship verification, while only the inputted watermarked samples have been changed to boundary samples during the verification phase. An important issue that should be considered is that these triggers have very low stealthiness. The effectiveness of this method would be significantly degraded if malicious users use Sota defense methods when training their models, which has already been demonstrated by current backdoor defense methods [1, 2].
> > >
> > > **R1**: Thank you for these comments and we do understand your concerns. We hereby provide more explanations to clarify potential misunderstandings that our submission and rebuttal may lead to you.
> > >
> > > - **ZeroMark focused only on the verification process of DOV methods**. It can be used to improve existing DOV methods by **replacing their verification process with ours while keeping their original watermarking process**.
> > > - We argue that **all your mentioned issues**, including the stealthiness of the watermark in the protected dataset and whether it can be detected during the training process **are irrelevant to our approach** since they are due to the property of user-specified watermarks instead of the verification process.
> > > - **We validated the effectiveness of our approach on different types of watermarking methods, not just on these two most basic ones (i.e., BadNets and Blended)**.
> > >     - Specifically, we also evaluate our method on WaNet and Domain Watermark. **These watermarks are highly stealthy**. Specifically, WaNet exploited warping-based image modification for watermarking. Its watermarks are sample-specific and imperceptible; Domain watermark used small additive noise for watermarking. Its watermarks are sample-specific and imperceptible, and under the clean-label setting.
> > >     - We have also conducted evaluations on domain watermark, which is the first non-backdoor-based dataset watermarking. **This watermark naturally bypass the detection of backdoor defenses**. Please find more details in our R4.
> > >
> > >
> > > ---
> > > **Q2**: For the unclear boundary, an important and in-depth ablation study about the Q% has been lost.
> > >
> > > **R2**: Thank you for this constructive suggestion! We have included the ablation study about the Q in our Ablation Study (Figure 6). **The results show that our method has a promising cosine similarity with various Qs**.
> > >
> > > ---
> > >
> > > **Q3**: Keeping the correct explanation about existing works (e.g., Blended) and equations (e.g., Eq. (5)) is crucial. If you "never" implement Eq. (5) in your experiment, why do you list it in your paper?
> > >
> > >
> > > **R3**: Thank you for pointing it out!
> > > - We will clarify our settings of Blended in our revision, as we mentioned and promised in our previous rebuttal.
> > > - We are deeply sorry that our previous rebuttal may lead you to some misunderstandings. **What we meant before was that we didn't calculate the similarity between a sample and its gradient; this formula is just used to explain specifically how the similarity is calculated**. In our method, we calculate the similarity between the trigger and the boundary gradient, as in our Theorem 1 (Eq.(6)). If we don't introduce the formula for similarity first up front, writing it directly later would make Theorem 1 very lengthy and more difficult to understand.
> > >
> > >
> > > We will add more explanations near Eq.(5) to avoid potential misunderstandings in our revision.
> > >
> > >
> > > ---
> > > **Q4**: Domain Watermark is a backdoor-based DOV method that employs clean-label backdoor attacks to implement dataset protection.
> > >
> > >
> > > **R4**: Thank you for this comment. However, as you may have misremembered or confused it with another work, **domain watermark is the first non-backdoor-based dataset watermark and DOV method**. It is clearly stated in its [original paper](https://arxiv.org/pdf/2310.14942) (Page 2, the second contribution). Specifically, backdoor-based methods made the watermarked model misclassify ‘easy’ samples that can be correctly predicted by the benign model. In contrast, domain watermark intended to make the watermarked model can correctly classify some 'hard' samples that will be misclassified by the benign model.
> > >
> > > We will add more details and explanations in the related works of our revision.
> > >
> > > ---

---

> > > > ### Author Response · Authors · 2024-08-09
> > > > **Responses to Post-rebuttal Comments (Round I) [Part 2]**
> > > >
> > > > **Q5**: Please don't over-claim your research topic. Unlearnable examples are a more feasible way to prevent unauthorized dataset usage than ZeorMark. The ZeroMark can only check whether someone has trained their models on your released dataset. If you check it, unauthorized dataset usage has already occurred.
> > > >
> > > >
> > > > **R5**: Thank you for your remind! We are deeply sorry that our previous rebuttal may have led you to some misunderstandings that we want to clarify here.
> > > >
> > > > - **We did not intend to claim that dataset ownership verification is a more important or feasible area than unlearnable samples**.
> > > > - As you mentioned, unlearnable examples and dataset ownership verification are active and passive defenses regarding preventing unauthorized dataset usage, respectively. They have different settings and usage scenarios.
> > > > - While dataset ownership verification is an after-the-fact method that does not directly prevent unauthorized use of a dataset, **DOV is, unfortunately, the only way in some scenarios (such as the one that is the focus of this paper)**. Specifically, when releasing open-sourced datasets (e.g., ImageNet) and selling commercial datasets, **we need to ensure that the datasets are available without compromising utilities** (and, therefore, cannot use unlearnable examples methods).
> > > >
> > > >
> > > > We will add unlearnable examples in our related work and provide more discussions and explanations in our revision.
> > > >
> > > > ---

---

> ### Author Response · Authors · 2024-08-10
> **Thanks to Reviewer s8Mh**
>
> Please allow us to thank you again for reviewing our paper and the valuable feedback, and in particular for recognizing the strengths of our paper in terms of *good soundness and presentation*, *extensive experiments*, and *novelty*.
>
> We also sincerely thank you for your timely follow-up feedback. Please kindly let us know if our response have properly addressed your remaining concerns. We are more than happy to answer any additional questions during the post-rebuttal discussion period. Your feedback will be greatly appreciated.

---

> > ### Comment · Reviewer_s8Mh · 2024-08-12
> > **Response to authors.**
> >
> > Thanks for your response. I haven't recognized the strong novelty of ZeroMark ever instead of concerning this issue. Detailed reasons are as follows:
> > 1. ZeroMark is indeed a verification method, however, its effectiveness is restricted to the backdoor attack methods that are employed. If malicious users adopt defense methods during the model training process, then ZeroMark cannot verify the copyright successfully, resulting in low practicality.
> > 2. The main idea of ZeroMark is to employ the reconstructed backdoored samples (i.e., boundary samples) to activate the injected backdoor. This is a straightforward application of backdoor defense, such as using backdoor defense methods to reconstruct the backdoored samples first and then to initiate backdoor attacks. The important boundary samples are generated by the existing method, FAB.
> >
> > Please find each reviewer correctly and respond to their comments. Thank you for understanding and I hope this will help improve your work.

---

> ### Author Response · Authors · 2024-08-12
> **A Gentle Reminder of the Post-rebuttal Discussion**
>
> Dear Reviewer s8Mh,
>
> We would like to sincerely thank you for your helpful comments. We hope our response has adequately addressed your concerns. We take this as a great opportunity to improve our work. We would be very grateful if you could kindly give any feedback to our rebuttal :)
>
> Best Regard,
>
> Paper4797 Author(s)

---

> ### Author Response · Authors · 2024-08-12
> **Responses to Post-rebuttal Comments (Round II)**
>
> Dear Reviewer s8Mh,
>
> Thank you for your follow-up comments. We greatly appreciate your efforts to help us improve our paper. However, we find that there are still some potential misunderstandings that may significantly mislead you. We hereby provide more explanations to address them.
>
>
> ---
> **Q1**: ZeroMark is indeed a verification method, however, its effectiveness is restricted to the backdoor attack methods that are employed. If malicious users adopt defense methods during the model training process, then ZeroMark cannot verify the copyright successfully, resulting in low practicality.
>
> **R1**: Thank you for your comments. We are deeply sorry that our previous rebuttal may have led you to some misunderstandings that we want to clarify here.
>
> - As we have mentioned before, **our ZeroMark method is not restricted only to backdoor-based watermarks**. For example, it can also be used to Domain Watermark, which is non-backdoor-based.
> - To the best of our knowledge, **there is no training-phase backdoor defense that can simultaneously defend against all types of backdoor attacks**. As such, even just for backdoor watermarks, we can't assume that users have been able to easily remove them completely through existing backdoor defenses.
> - We argue that **it is unfair to completely dismiss our work simply because baseline watermarks may be removed**. This is just a probability and it is out the scope of this paper (PS: it is about the robustness of dataset watermarks).
>
>
> ---
>
> **Q2**: The main idea of ZeroMark is to employ the reconstructed backdoored samples (i.e., boundary samples) to activate the injected backdoor. This is a straightforward application of backdoor defense, such as using backdoor defense methods to reconstruct the backdoored samples first and then to initiate backdoor attacks. The important boundary samples are generated by the existing method, FAB.
>
> **R2**: Thank you for your comments. We are deeply sorry that our previous rebuttal may have led you to some misunderstandings that we want to clarify here.
> - **Boundary samples are not backdoor samples**, as shown in its definition (Eq.(1)&Eq.(3)). As such, **our ZeroMark does not have any direct relation to the backdoor sample synthesis (or backdoor trigger inversion)**.
>
> ---
> **The Definition of Boundary Samples**. Let the logit margin of model $f: \mathcal{X} \rightarrow [0,1]^K$ on the label $y$ is denoted by:
> \begin{align}
>     \phi_{y}(x;w) = f_{y}(x;w) - \max_{y'\neq y} f_{y'}(x;w).
> \end{align}
> It can be observed that $x$ can be classified as $y$ by $f(\cdot;w)$ if and only if $\phi_{y}(x;w) \geq 0$. As such, the set for boundary samples of class $y$ can be denoted by $\mathcal{B}(y;w) = \{x: \phi_{y}(x;w) = 0\}$.
>
> ---
>
> - **The core contribution of the paper is not how to compute the boundary samples**, but rather the definition of this new research problem and our empirical and theoretical findings of the intrinsic property of DNNs trained on the watermarked dataset.
> - We argue that **it is unfair to completely dismiss our work simply because we need to use existing techniques to calculate some intermediate results**. Otherwise, we can easily infer that almost all existing DL works has limited novelty, as they all require updating the model with PGD/SGD, etc.
>
>
>
> ---
>
> **Q3**: Please find each reviewer correctly and respond to their comments.
>
> **R3**: Thank you for your comments. You may have this misunderstanding because we summarized the key points of your original question in our rebuttal due to word limitations. Although we have answered your original questions in the order they were asked and have tried to maintain enough key information, we apologize for any misunderstandings we may have caused you.
>
> Please kindly let us know if we missed any points. We are very happy to answer them before the discussion period ends.
>
>
>
> ---

---

### Official Review · Reviewer_RJcM · 2024-07-09

**Soundness:** 3
**Presentation:** 3
**Contribution:** 3
**Rating:** 5
**Confidence:** 4

**Summary:**

This work presents a method for dataset ownership verification, focusing on confidentiality during the verification phase. The authors highlight that adversaries can remove watermarked data by detecting it during verification. To address this issue, the paper proposes a new verification process based on the cosine similarity between the watermark pattern and the gradient at boundary samples. Building upon their observations and Theorem 1, the authors use the similarity between these two components as evidence of unauthorized dataset use.

**Strengths:**

This work separates the verification sample from the watermarked samples. By doing so, it is possible to protect the watermarked samples from adversaries.

**Weaknesses:**

However, I have some concerns as follows:

1) The authors state that "Theorem 1 indicates that the cosine similarity between the watermark pattern and the gradient increases along with the update process." However, I'm not convinced. As I understand it, cosine similarity must be within the range [-1, 1], so the left term in Eq (6) may be within [0, 2]. Then,
t* represents the number of iterations, and the authors used 10-40 iterations as shown in Figure 5 (always a positive integer). This implies a simple condition where c>0. In this case, I question whether the lengthy proof is necessary.


2) Furthermore, Theorem 1 does not imply a proportional relationship between the two terms. It suggests Lipschitz continuity, not a positively increasing relationship. Therefore, I am not convinced about the reliability of the proposed verification method.


3) Furthermore, I find this method to be rather impractical. The proposed approach necessitates the predicted logits for generating boundary data and conducting verification. However, full logits are generally not available in commercial services. This requirement appears impractical to me. In contrast, other methods, such as BadNet or Blended, can perform verification using only the one-hot encoded predictions, indicating the Top-1 prediction.  Although they can verify using only Top-1 predictions, all results in this paper rely on the logits. For a fair comparison, I believe the authors should compare their method with others that also use logits for verification.


4) I find the experiments (e.g., ResNet/VGG on CIFAR10/TinyImageNet without error bars) insufficiently convincing. In addition, the authors mention, "We have reported the distribution of our results in Figure 2, and Appendix B" at the checklist, but they only describe histograms of 400 samples. I don't consider the histograms to be a substitute for error bars because the results are achieved from a single model, which must represent reliability through multiple suspicious/clean models.


5) The proposed method primarily uses backdoor attacks with label noise as a baseline. However, there are clean-labeled verification approaches, such as radioactive data [1]. I believe the property of clean labels is highly important in dataset watermarking, but I'm unsure whether the proposed method can work with clean-labeled data.


[1] Sablayrolles, A., Douze, M., Schmid, C., & Jégou, H. (2020, November). Radioactive data: tracing through training. In International Conference on Machine Learning (pp. 8326-8335). PMLR.

**Questions:**

1) For the confidentiality (or security) of the proposed method, isn't it possible to reconstruct the watermark (BadNet, Blended, and WaNet) using multiple boundary samples? According to Figure 4, the proposed verification requires multiple gradients for these boundary samples. If I were the adversary, I might attempt to reconstruct the watermark pattern using the multiple boundary samples and their corresponding gradients.

2) I expect that the proposed method is affected by data augmentation techniques such as Flip, MixUp, or random noise injection. Additionally, I think the proposed method can be compromised by random dropout. For example, if random flip was applied during training of the suspicious model, the boundary gradient would become less correlated to the watermark pattern because the model would have also seen its mirror pattern.

**Limitations:**

As I mentioned, the authors doesn't represent the error bars.

---

> ### Author Rebuttal · Authors · 2024-08-06
>
> Dear Reviewer RJcM, thank you very much for your careful review of our paper and thoughtful comments. We are encouraged by your positive comments on our **good presentation** and **novel verification process**. We hope the following responses can help clarify potential misunderstandings and alleviate your concerns.
>
> ---
>
> **Q1**: I question whether the lengthy proof of Theorem 1 is necessary.
>
>
> **R1**: Thank you for your insightful comment!
>
> - **The right term in Eq (6) decrease instead of increase with the increase of the number of iterations $t^{*}$**. As such, **this inequality is not naturally hold** even if the left size may be within [0, 2] and $c$ could be large.
> - **This potential misunderstanding may due to the format of the right side**, i.e., $c\cdot (t^*)^{q-1}$ where $q\in(\frac{1}{2},1)$. To avoid this misunderstanding, we can rewrite it as: $c \cdot \frac{1}{(t^{*})^{b}}$, where $b = 1-q \in (0,\frac{1}{2})$.
> - **We have also empirically verified it in Figure 5**. As shown in this Figure, the cosine similarity scores increase with the increase of $t$.
>
>
> We will add more details and discussions in our revision.
>
>
> ---
>
> **Q2**: Theorem 1 suggests Lipschitz continuity, not a positively increasing relationship.
>
> **R2**: Thank you for your insightful comment!
>
> - As we mentioned in R1, we can rewrite the right side of Theorem 1. **The new format can clearly suggest a a positively increasing relationship**.
> - **We have also empirically verified it in Figure 5**. As shown in this Figure, the cosine similarity scores increase with the increase of $t$.
> - Arguably, **the inequality does not have much to do with Lipschitz continuity** since the right size is not about the rate of change of a variable.
>
> We will add more details and discussions in our revision.
>
> ---
>
> **Q3**: Zeromark needs the predicted logits for conducting verification.
>
> **R3**: Thank you for your insightful comment!
>
> - **Our approach does not require logits for generating boundary samples**. The potential misunderstanding may come from Eq.(10). But our gradient estimation and generation processes are consistent with previous black-box adversary attacks under the hard-label setting.
> - In our released zeromark.py, **Line 123-140 (function for obtaining the final predictive label), Line 194-238 (gradient estimation) and Line 253-293 (geometric search for boundary samples)** verify that ZeroMark is implemented requiring only predicted labels for inputs.
>
> We will add more details in our revision.
>
> ---
>
> **Q4**: The authors should provide error bars.
>
> **R4**: Thanks for your constructive suggestion! Following your suggestion, we evaluate our main experiments with four independent models and report the error bars. As shown in Table 2-4 in our uploaded PDF, **the results are sufficiently consistent with a low std**.
>
> ---
>
> **Q5**: Whether the proposed method can work with clean-labeled data.
>
> **R5**: Thanks for your insightful comment! We agree with you that  clean label is highly importatnt in dataset watermarking.
>
>
> - **ZeroMark can perform effective with the most recent clean-label watermark** (i.e., Domain Watermark).
> - Following your suggestion, we have tried to combine our method to radioactive data.
>     - However, **it is ineffective and impractical for dataset ownership verification under our considered threat model with a small watermarking rate**. We find the gap for watermark inputs yields nearly the same cross-entropy loss as benign inputs with AUROC as 0.507.
>     - It requires to calculate the cross entropy, needing all logits.
>
> ---
>
> **Q6**: Reconstruct the watermark pattern using the multiple boundary samples and their corresponding gradients.
>
> **R6**: Thanks for this insightful comment!
>
> - In general, **ZeroMark is resillient against watermark reconstruction**. Due to the page limitation, we have conducted and placed this experiment in Appendix J (Figure 21-22).
> - In our experiments, we used domain watermark as an example since it is the most robust and SOTA dataset watermark. To further alleviate your concern, we evaluate ZeroMark also on BadNet, Blended, and WaNet. As shown in Figure 2 in the uploaded PDF, **the adversaries cannot reconstruct the watermark pattern via boundary samples**.
> - Arguably, the reconstruction failure is mostly because
>     - **The boundary samples contain limited information of the trigger patterns**, as shown in Table 2-3.
>     - **Boundary samples generated by ZeroMark can always stay far away from the watermark samples' distribution** (as shown in Figure 9). It demonstrates ZeroMark can prevent disclosing the watermark information from the watermark samples.
>
>
> We will provide more discussions in our revision and further explore it in our future works.
>
> ---
>
> **Q7**: Effects for the data augmentation techniques (random flip) and dropout during the training phase of suspicious model.
>
> **R7**: Thanks for your construstive suggestions!
>
> -- **ZeroMark is resillient to data augmentation and random dropout during the training phase.** We have already trained all evaluated models with data augmentation tenciques and random dropout. Our results show that ZeroMark performs resillient against the data augmentation techniques.
> -- **Random Flip does not cause "mirror patterns" of watermark for suspicious model**.
>  - Following your suggestion, we evaluate the suspicious model built with RandomHorizonFlip against inputs attached with the watermark pattern and its mirror pattern for BadNets on CIFAR-10. We find that **the suspicious model can achieve a 99.9% VSR on inputs containing watermark but only a 9.95% VSR on inputs with the mirror one**.
>  - It indicates the **random flip will not affect the boundary gradient correlated to the original watermark pattern**.
>  - We speculate it mostly because the watermark pattern correlated to the semantic feature of inputs rather than the relative position.
>
> We will add more details and discussions in the appendix of our revision.
>
> ---

---

> ### Author Response · Authors · 2024-08-10
> **Thanks to Reviewer RJcM**
>
> Please allow us to thank you again for reviewing our paper and the valuable feedback, and in particular for recognizing the strengths of our paper in terms of *good presentation* and *novel verification process*.
>
> Kindly let us know if our response and the new experiments have properly addressed your concerns. We are more than happy to answer any additional questions during the post-rebuttal discussion period. Your feedback will be greatly appreciated.

---

> > ### Comment · Reviewer_RJcM · 2024-08-11
> >
> > Thank you for the thoughtful reply.
> >
> > First, I agree on the importance of protecting the verification data, and I acknowledge that the proposed method addresses this problem effectively. Then, thanks for this new aspect of protection.
> >
> > Regarding Theorem 1, I apologize for my earlier misunderstanding. I now understand that the right term decreases as $t^*$ increases.
> >
> > For the experiments, thanks for the additional results and explanations. They help me to understand the propose method. However, I feel that the limited settings (CIFAR-10 and Tiny ImageNet with ResNet and VGGNet) are not sufficient to concretely verify general applicability. I believe the evaluation could be more robust if more diverse settings were included. For example, I am curious about the method's applicability to clean-labeled backdoor attacks, such as Sleeper Agent.
> >
> > Many of my concerns have been addressed, so I have raised my rating.

---

> > > ### Author Response · Authors · 2024-08-12
> > > **Responses to Post-rebuttal Comments (Round 1) [Part 1]**
> > >
> > > Dear Reviewer s8Mh,
> > >
> > > We sincerely thank you for your timely follow-up feedback! We are so glad that our previous responses clarified potential misunderstandings and alleviated your concerns to a large extent. We also deeply thank you for your positive feedback, especially comments regarding our research importance, method effectiveness, and a new protection aspect. It encourages us a lot! We promise to add more discussions and experiments that we previously committed to in the revision.
> > >
> > > In this response letter, we hope to further alleviate your remain concerns. We believe this can further improve our work. Thank you again for giving us this chance :)
> > >
> > > ---
> > > **Q1**: I feel that the limited settings (CIFAR-10 and Tiny ImageNet with ResNet and VGGNet) are not sufficient to concretely verify general applicability.
> > >
> > > **R1**: Thank you for these insightful comments! We do agree with you that generalizability and flexibility across models and datasets are also important for the the general applicability of a method. We hereby provide more explanations and results to further alleviate your concerns.
> > >
> > > - **Generalizability to Other Model Architectures** (e.g., Transformer).
> > >     - In general, the success of our approach on other model structures depends on two factors: **(1)** whether the studied dataset watermarking method (e.g., BadNets) can successfully watermark these models and **(2)** whether we can conduct effective 'adversarial attacks' to find boundary samples on these models. Based on existing work related to backdoor attacks/dataset watermarking [1,2] and adversarial attacks [3,4], **these factors are all met**. As such, **our method can fundamentally generalize to other models (e.g., transformer) as well**.
> > >     - To further alleviate your concern, as you suggested, we hereby also evaluate our ZeroMark on the transformer archietecture. We empirically evaluate the effectiveness of ZeroMark on the TinyImageNet dataset using SwinTransformer with a patch size of 4. Other settings are consistent with our main experiments. **As shown in the following Table 1-2, our method is highly effective under SwinTransformer**.
> > >
> > > **Table 1.** The top $Q\%$ cosine similarity scores of ZeroMark on Tiny-ImageNet with SwinTransformer.
> > >
> > > | Label$\downarrow$, Dataset$\rightarrow$ | BadNets | Blended | WaNet | DW    |
> > > |----------------|---------|---------|-------|-------|
> > > | Benign         | 0.034   | 0.034   | 0.039 | 0.041 |
> > > | Target         | 0.096   | 0.221   | 0.124 | 0.119 |
> > >
> > >
> > > **Table 2.** The verification efficacy of ZeroMark with SwinTransformer.
> > >
> > > | Watermark$\downarrow$ | Scenario$\rightarrow$   | Independent-D | Independent-M | Malicious |
> > > |-----------|------------|---------------|---------------|-----------|
> > > | BadNets   | $\Delta P$ / p-value |     0.010 / 1.00              |  0.011  / 1.00           |      0.062  / $10^{-9}$   |
> > > | Blended   |     $\Delta P$ / p-value       |    0.011  / 1.00         |     0.014   / 1.00       |  0.187 / $10^{-61}$         |
> > > | WaNet     |    $\Delta P$ / p-value        |  0.017 / 0.99             |     0.011 / 1.00           |   0.085 / $10^{-46}$        |
> > > | DW        |     $\Delta P$ / p-value       | 0.021 / 0.90              |  0.010 / 1.00             |        0.078 / $10^{-19}$   |
> > >
> > > - **Generalizability to Other Datasets** (e.g., NLP Datasets with Discrete Data Format).
> > >     - **We conduct experiments only on these two image datasets simply following existing works** [2, 5]. Due to the limitation of paper length, it is also difficult for us to provide a comprehensive evaluation on other data types.
> > >     - However, we do understand your concern. Arguably, the main challenge lies in how to design effective adversarial attacks to discrete data formats for finding the closest boundary samples (as in Eq.(10)). In particular, there are already some relevant works [6, 7] confirming its feasibility. Accordingly, **our ZeroMark can be naturally adapted to other discrete data formats** (e.g., tabular or text). Due to the limitation of time, we cannot provide sufficient experiments to verify it in our rebuttal. We will further discuss it in our future works.
> > >
> > > ### References
> > >
> > > 1. You Are Catching My Attention: Are Vision Transformers Bad Learners under Backdoor Attacks? CVPR, 2023.
> > > 2. Domain Watermark: Effective and Harmless Dataset Copyright Protection is Closed at Hand. NeurIPS, 2023.
> > > 3. On the robustness of vision transformers to adversarial examples. ICCV, 2021.
> > > 4. On the Adversarial Robustness of Vision Transformers. TMLR, 2022.
> > > 5. Untargeted Backdoor Watermark: Towards Harmless and Stealthy Dataset Copyright Protection. NeurIPS, 2022.
> > > 6. TextCheater: A Query-Efficient Textual Adversarial Attack in the Hard-Label Setting. TDSC, 2023.
> > > 7. Query-Efficient and Scalable Black-Box Adversarial Attacks on Discrete Sequential Data via Bayesian Optimization. ICML, 2022.
> > >
> > > ---

---

> > > > ### Author Response · Authors · 2024-08-12
> > > > **Responses to Post-rebuttal Comments (Round 1) [Part 2]**
> > > >
> > > > ---
> > > > **Q2**: I am curious about the method's applicability to clean-labeled backdoor attacks, such as Sleeper Agent.
> > > >
> > > > **R2**: Thank you for this insightful question! We do understand your concern since we did not evaluate our method on clean-label backdoor watermarks, although we have evaluated our ZeroMark on the SOTA non-backdoor-based clean-label watermarks (i.e., DW). We hereby provide more explanations and results to further alleviate your concerns.
> > > >
> > > > - In principle, **our ZeroMark can be used for all backdoor watermarks, regardless of whether they have dirty-label or clean-label settings**, since our work targets the inference instead of the training process.
> > > >     - **The predictive behavior of both types of attacks is consistent**, i.e., making the attacked models have a specific prediction (i.e., target label) whenever the trigger pattern appears.
> > > >     - **Both types of attacks lead to similar changes in the feature space**: the poisoned samples are clustered into a single cluster and moved away from the respective clusters of the other types of benign samples.
> > > >     - **Both types of attacks lead to changes in the decision surface that are highly correlated with triggers**, since both cause samples with triggers to be misclassified, of attacked models.
> > > >     - **Our Zeromark makes no assumptions about the label type**.
> > > > - To further alleviate your concern, **we conduct additional experiments on CIFAR-10 with label-consistent attack and SleeperAgent**. They are the first and the advanced clean-label backdoor watermark, respectively.
> > > >     - As shown in the following Table 3-4, **our method is still effective under clean-label backdoor watermarks**.
> > > >     - It is worth mentioning that our method requires that models already learn the watermark, which may require a larger watermarking rate for clean-label backdoor watermarks. However, **this problem stems from the intrinsic properties of these watermarks** (e.g., the antagonistic effects of 'robust features' [3]) **and is irrelevant to our ZeroMark**.
> > > >
> > > > **Table 3.** The top $Q\%$ cosine similarity scores of ZeroMark on CIFAR-10 with ResNet-18.
> > > >
> > > > | Label$\downarrow$, Dataset$\rightarrow$ | Sleeper Agent [1]| Label-Consistent [2] |
> > > > |----------------|---------|---------|
> > > > | Benign         |  0.033   |0.029   |
> > > > | Target         | 0.120  |  0.104   |
> > > >
> > > >
> > > >
> > > > **Table 4.** The verification efficacy of ZeroMark with label-consistent attack and SleeperAgent.
> > > >
> > > > | Watermark$\downarrow$ | Scenario$\rightarrow$   | Independent-D | Independent-M | Malicious |
> > > > |-----------|------------|---------------|---------------|-----------|
> > > > | Sleeper Agent [1]  |     $\Delta P$ / p-value       |    0.012  / 1.00         |     0.010   / 1.00       |  0.087 / $10^{-48}$         |
> > > > | Label-Consistent [2]  | $\Delta P$ / p-value |     0.014 / 1.00              |  0.009  / 1.00           |      0.075  / $10^{-15}$   |
> > > >
> > > >
> > > > We will add more details and discussions in the appendix of our revision.
> > > >
> > > >
> > > >
> > > > ### References
> > > >
> > > > 1. Sleeper Agent: Scalable Hidden Trigger Backdoors for Neural Networks Trained from Scratch. NeurIPS, 2022.
> > > > 2. Label-Consistent Backdoor Attacks. arXiv, 2019.
> > > > 3. Not All Samples Are Born Equal: Towards Effective Clean-Label Backdoor Attacks. Pattern Recognition, 2023.
> > > >
> > > >
> > > > ---

---

### Official Review · Reviewer_RHDz · 2024-07-11

**Soundness:** 3
**Presentation:** 3
**Contribution:** 3
**Rating:** 7
**Confidence:** 4

**Summary:**

This paper explores how to conduct privacy-preserving dataset ownership verification without directly disclosing dataset watermarks. The proposed method is inspired by the characteristic of boundary gradient of watermarked DNNs. Specifically, it has three main steps, including (1) generate the (closest) boundary samples, (2) calculate the boundary gradients of the generated boundary samples, and (3) dataset ownership verification via boundary gradient analysis. The authors conduct experiments on CIFAR-10 and Tiny-ImageNet datasets with four baseline methods.

**Strengths:**

1.	Dataset copyright protection is of great significance and sufficient interest to NeurIPS audiences. In particular, this paper explores a new angle in dataset ownership verification and studies the verification process for the first time. I think it enlightens this area and can inspire follow-up research. Conduct verification without disclosing the secret key (i.e., watermark) is of practical importance.
2.	The motivation section (Section 3.3) is interesting and insightful. The authors reveal an intriguing phenomenon and also provide its theoretical analysis. I enjoy reading this part.
3.	The proposed method is simple yet effective. It can also be used to improve different types of existing watermarking methods.
4.	The authors have provided their codes. It should be encouraged.

In general, I think this is a good paper with deep insights and good performance. However, I still have some concerns and questions that could help to improve this paper further, as follows.

**Weaknesses:**

1.	The motivation section is inspiring. However, the authors should provide more explanations about how they came up with this idea and how to further explore it. It is crucial to highlight follow-up research.
2.	The authors should justify the phenomenon in Figure 2 is not exists in benign models.
3.	It would be better to explain more about how Theorem 1 relates to the proposed method.
4.	The authors also used BadNets in the experiments of Section 3.3. It would be better to also show results of other watermarking methods, especially those with non-patch-based watermarks.
5.	The authors introduce a normalization process in the calculation of cosine similarity score. The authors should conduct ablation study to verify its effectiveness.
6.	Missing setting details used for the zero-order gradient estimation (i.e., N and \beta_t).
7.	The authors should also analyze the visual similarities between the optimized verification samples and their benign version, as well as with the ground-truth dataset watermark, w.r.t. to the optimization process.
8.	The authors should also conduct experiments on methods with non-compact watermarks.
9.	The authors should provide more explanations in Section 5.5.

There are still some typos. For example, ‘method’ should be ‘model’ (Line 16, Page 1).

**Questions:**

Please see the weaknesses part.

**Limitations:**

Yes

---

> ### Author Rebuttal · Authors · 2024-08-06
>
> Dear Reviewer RHDz, thank you very much for your careful review of our paper and thoughtful comments. We are encouraged by your positive comments on our **great significance**, **intriguing phenomenon with theoretical analysis**, **novel and interesting method**, **simple and effective method**,, **good presentation**, and **good soundness**. We hope the following responses can help clarify potential misunderstandings and alleviate your concerns.
>
>
>
> ---
> **Q1**: More explanations about the motivation of ZeroMark?
>
> **R1**: Thank you for this insightful question! In general, our approach is **motivated by previous works on model fingerprints to some extent, where we can use decision boundary properties to attribute DNNs**. We think the trigger pattern is simply a straightforward method to probe the decision boundary of the suspicious model.
>
> ---
>
> **Q2**: The authors should justify the phenomenon in Figure 2 is not exists in benign models.
>
> **R2**: Thank your for this contrusctive suggestion! During the rebuttal period, we plot the distribution for cosine similarity under benign models on the Tiny ImageNet dataset in our loaded PDF (Figure 1). It shows that **this phenomenon does not exist in benign models**.
>
> ---
>
> **Q3**: It would be better to explain more about how Theorem 1 relates to the proposed method.
>
> **R3**: Thank you for pointing it out! We are deeply sorry that our submission failed to provide sufficient information, which we want to clarify here.
> - Therom 1 discusses the error for the consine similarity computed over boundary samples and the corresponding watermark pattern during the optmization process of Eq.(4).
> - During the procedure of optimizing Eq.(4), when $t^{*}$ becomes large, **the error for the consine similarity computed over boundary samples and the corresponding watermark pattern gets smaller**.
> - Motivated by that, we **seek the closest boundary samples with a large $t^{*}$ to minimize the error** in our ZeroMark method.
>
> We will provide more details and discussions in our revision.
>
>
> ---
>
> **Q4**: It would be better have evaluation for non-patch-based watermarks.
>
> **R4**: Thank you for this insightful comment!
> - **We have evaluated ZeroMark for non-patch-based watermarks** (e.g., WaNet and Domain Watermark) in our experiments. Due to the space limitation, we put them in our appendix (Section C).
> - In general, **their results are consistent with those of BadNets**, i.e., this phenomenon is universal. We will add more details and discussions in our revision.
>
> We will provide more details and discussions in our revision.
>
> ---
>
> **Q5**: The effect for the normalization process.
>
> **R5**: Thank you for this insightful comment!
> - Following your suggestions, we conduct an ablations study on the TinyImageNet dataset to show the effectiveness of the normalization process. The results are shown in our uploaded PDF (Figure 1).
> - The results show that **it is hard to identify the benign and watermark models' distributions on cosine similarity without normalization**. It verifies the effectiveness of this process.
>
> We will provide more details and discussions in the appendix of our revision.
>
> ---
>
> **Q6**: Missing setting details used for the zero-order gradient estimation (i.e., N and \beta_t).
>
> **R6**: Thank you for pointing it out! We are deeply sorry that our submission failed to provide sufficient setting information, which we want to clarify here. Fllowing the classical zero-order optimization process, we set $N$ as 200. The $\beta_{t}$ is calcualted following Theorem 1. We will add more details in our revision.
>
> ---
>
> **Q7**: Analysis on the visual similarities between the optimized verification samples and their benign version, as well as with the ground-truth dataset watermark, w.r.t. to the optimization process.
>
> **R7**: Thanks for your constructive suggestions. We evaluate the visual similarities between optimized verification samples and their benign version and their benign version in our uploaded PDF. The results show that the visual similarity between boundary and benign samples increases and then becomes stable during the optimization procedure, indicating the stealthiness and non-disclosure property of our ZeroMark.
>
> ---
>
>
> **Q8**: The authors should also conduct experiments on methods with non-compact watermarks.
>
> **R8**: Thank you for this constructive suggestion! We are deeply sorry that our submission may lead you to some potential misunderstandings that we want to clarify here.
> - In our paper, **we have conducted experiments on non-compact wateramarks**, including WaNet and Domain Watermark. Their watermarks are all over the whole watermarked images.
> - As shown in Table 1-4 of our main manuscript, **our method is highly effective on them**.
>
> We will add more details in our revision.
>
> ---
>
>
> **Q9**: The authors should provide more explanations in Section 5.5.
>
> **R9**: Thank you for pointing it out! We are deeply sorry that our submission failed to provide sufficient explanations.
> - As ZeroMark obtained the closest boundary samples by optimizing Eq.(4), we thus evaluate the effectiveness of ZeroMark during the optimization procedure of Eq.(4) with varying steps $t$.
> - As shown in Figure 9, **ZeroMark is consistently separated from watermark samples during the optimization procedure of Eq.(4) with varying steps $t$**.
> - As such, ZeroMark can prevent the disclosure of watermark information from the watermark samples.
>
> We will add the details in the appendix of our revision.
>
>
> ---

---

> ### Comment · Reviewer_RHDz · 2024-08-09
>
> I would like to thank the authors for providing a detailed rebuttal containing additional experiments. It addressed all my concerns. Well done!
>
> I have also carefully read the comments from other reviewers, especially Reviewer RJcM and Reviewer s8Mh, since they have different judgments of this paper. After reading their comments and the rebuttal, I think their negative opinions are mostly due to misunderstandings, which are caused by the loss of some technical details or explanations. I believe the authors did a good job of clearing up these misunderstandings in their rebuttal. In addition, other expert reviewers raised some concerns that I had not thought of. But I think the authors also addressed them in the rebuttal, at least to me.
>
> Given all these considerations, as well as insightful motivation, novelty, and potential impact on this field, I slightly increase my score.  I would also be happy to hear from other reviewers and participate in discussions :)

---

> > ### Author Response · Authors · 2024-08-09
> > **Thank You for Your Positive Feedback!**
> >
> > Dear Reviewer RHDz,
> >
> > Thank you so much for your positive feedback! It encourages us a lot!
> >
> > We are also thrilled to answer all follow-up questions during the discussion period.

---

### Official Review · Reviewer_VV25 · 2024-07-13

**Soundness:** 3
**Presentation:** 2
**Contribution:** 3
**Rating:** 6
**Confidence:** 3

**Summary:**

This paper proposes ZeroMark, a novel scheme for dataset watermark verification. It is based on an observation that the boundary gradients (i.e., gradients of samples near the decision boundary of a watermarked model) of the watermark target class tend to have higher cosine similarities with the watermark pattern than the boundary gradients of benign classes. ZeroMark hence proposes to verify the watermark based on the cosine similarities between the boundary gradients and the watermark pattern. In this way the actual watermark pattern is not disclosed to the adversary during verification, thus making watermark verification more practical as it prevents the adversary from exploiting the leaked watermark pattern. Experimental results show that ZeroMark leaks little information about the original watermark pattern, achieves high verification performance and is relatively robust against removal attempts.

**Strengths:**

1. It addresses an interesting and relatively new problem in dataset watermarking. Protecting the watermark pattern during watermark verification is beneficial since it makes it more difficult for the adversary to infer the watermark pattern and remove the watermark.
2. It proposes an interesting observation. The observation that the boundary gradients of the target class tend to have higher cosine similarities with the watermark pattern could be potentially beneficial for future research on watermarking or backdooring.
3. The proposed method could work in conjunction with various existing watermarking techniques.
4. This work contains rather solid experimental evaluation.

**Weaknesses:**

1. The proposed method is primarily evaluated on image datasets and convolution networks (ResNet and VGG) and lacks a discussion on the potential applicability to other data formats (e.g., tabular or text) or model architectures (e.g., transformers).
2. The adaptive attacks discussed in the appendix lack details on experimental settings.
3. Some parts of the presentation in the manuscript are confusing or unclear.

**Questions:**

1. (Applicability to other data formats or model architectures) The evaluation on image datasets is quite comprehensive. However, it would be better if the authors could add a discussion on how ZeroMark extends to other data formats or model architectures. For example, for text data, it might be difficult to construct boundary examples using Eq.10 since texts are discrete.
2. (Adaptive attack setting) The paper mentions two adaptive attacks in appendix J. However, the appendix lacks some details for the adversary's setup (e.g., steps for recovering the watermark pattern, or hyper-parameters for unlearning the watermark).
3. (Clarity) The "largest Q% cosine similarity" used in the experiment is confusing: does this Q has a specific value in the experiments?
4. (Clarity) How the cosine similarity is computed seems confusing. In Eq.29 in the appendix, the cosine similarity is computed on the watermark patch, identified by a "location map $m$". However, this location map is not mentioned in Eq.8 in the main body.
5. (Clarity) Fig.4 presents 5 different samples constructed by ZeroMark, but it lacks an explanation on the differences among the 5 samples.

**Limitations:**

The authors have acknowledged some limitations of ZeroMark in Appendix K, including (1) additional time overhead due to the construction of boundary samples and (2) theoretical guarantee on the security of ZeroMark.

---

> ### Author Rebuttal · Authors · 2024-08-06
>
> Dear Reviewer VV25, thank you very much for your careful review of our paper and thoughtful comments. We are encouraged by your positive comments on our **novel and interesting research problem**, **interesting observation**, **method flexibility**, and **extensive and solid experiments**. We hope the following responses can help clarify potential misunderstandings and alleviate your concerns.
>
> ---
> **Q1**: It would be better to add a discussion on how ZeroMark extends to other data formats or model architectures.
>
> **R1**: Thank you for these insightful comments! We do agree with you that generalizability and flexibility are also important for a method.
> - **Generalizability to Other Model Architectures**.
>     - In general, the success of our approach on other model structures depends on two factors: **(1)** whether the studied dataset watermarking method (e.g., BadNets) can successfully watermark these models and **(2)** whether we can conduct effective 'adversarial attacks' to find boundary samples on these models. Based on existing work related to backdoor attacks/dataset watermarking [1,2] and adversarial attacks [3,4], **these factors are all met**. As such, **our method can fundamentally generalize to other models (e.g., transformer) as well**.
>     - To further alleviate your concern, as you suggested, we hereby also evaluate our ZeroMark on the transformer archietecture. We empirically evaluate the effectiveness of ZeroMark on the TinyImageNet dataset using SwinTransformer with a patch size of 4. Other settings are consistent with our main experiments. **As shown in Table 1 in our uploaded PDF, our method is highly effective with verification efficacy**.
> - **Generalizability to Other (Discrete) Data Formats**.
>     - **We conduct experiments only on image datasets simply following existing works** [2, 5]. Due to the limitation of paper length, it is also difficult for us to provide a comprehensive evaluation on other data types.
>     - However, we do understand your concern. Arguably, the main challenge lies in how to design effective adversarial attacks to discrete data formats for finding the closest boundary samples (as in Eq.(10)). In particular, there are already some relevant works [6, 7] confirming its feasibility. Accordingly, **our ZeroMark can be naturally adapted to other discrete data formats** (e.g., tabular or text). Due to the limitation of time, we cannot provide sufficient experiments to verify it in our rebuttal. We will further discuss it in our future works.
> ## Ref
> 1. You Are Catching My Attention: Are Vision Transformers Bad Learners under Backdoor Attacks?
> 2. Domain Watermark: Effective and Harmless Dataset Copyright Protection is Closed at Hand.
> 3. On the robustness of vision transformers to adversarial examples.
> 4. On the Adversarial Robustness of Vision Transformers.
> 5. Untargeted Backdoor Watermark: Towards Harmless and Stealthy Dataset Copyright Protection.
> 6. TextCheater: A Query-Efficient Textual Adversarial Attack in the Hard-Label Setting.
> 7. Query-Efficient and Scalable Black-Box Adversarial Attacks on Discrete Sequential Data via Bayesian Optimization.
>
> ---
> **Q2**: Lacks some details for the adversary's setup (e.g., steps for recovering the watermark pattern, or hyper-parameters for unlearning the watermark).
>
> **R2**: Thank you for pointing it out! We are deeply sorry that our submission failed to provide sufficient setting information that we want to clarify here.
> - **Setups for Recovering the Watermark Pattern**. As ZeroMark sends several (e.g., 200) boundary samples attached with random perturbations $\lbrace\bar{x}+\mu_{I}\rbrace_{i=1}^{n}$ for gradient estimation purposes, the adversary would aggregate the queried boundary samples $\lbrace\bar{x}+\mu_{I}\rbrace_{i=1}^{n}$ to recover the trigger pattern following $\frac{1}{n}\sum_{i=1}^{n} (\bar{x}+\mu_{i})$.
> - **Setups for Machine Unlearning**. For a watermarked model $f(\cdot;\theta)$, we fine-tune the watermarked model with collected boundary samples $\lbrace\bar{x}+\mu_{I}\rbrace_{i=1}^{n}$ labeling with its ground truth label $y$. Specifically, we unlearn the watermarked model following: min $E_{\bar{x}} [\frac{1}{n}\sum_{i=1}^{n}\ell(f(\bar{x}+\mu_{i};\theta),y)]$. We collect 500 pairs of boundary samples for unleanring. The learning rate is set as 0.001 and we fine-tune the model 100 epochs.
>
> ---
> **Q3**: Does this Q has a specific value in the experiments?
>
> **R3**: Thank you for pointing it out! We are deeply sorry that our submission failed to provide sufficient setting information, which we want to clarify here. We **set Q as 10** in our experiments (Section 5.1-5.2), i.e., we select the largest 10% cosine similarity out of m (500) cosine similarity to perform a t-test. We have also performed an ablation study to understand the performance with varying Q in Section 5.3 (Figure 6). The results show that **our method can still have a promising cosine similarity with various Qs**.
>
> ---
> **Q4** How the cosine similarity is computed seems confusing.
>
> **R4**: We are deeply sorry that our submission may lead you to some misunderstandings that we want to clarify. We compute the cosine similarity on the watermark pattern $\delta$ and the estimated gradients **located within the watermark pattern's region**. We will clarify and detail this process in our revision.
>
> ---
> **Q5** What's the differences among the 5 samples in Fig.4 for ZeroMark.
>
> **R5**: Thank you for pointing it out! We are deeply sorry that our submission failed to provide sufficient information, which we want to clarify here.
> - The five ZeroMark samples in Fig.4 are the **boundary samples with different optimization iterations** for a given sample during the procedure of optimizing Eq.(10).
> - Notably, **ZeroMark uses all these figures to query the suspicious model**, but only the last one is the closet boundary sample.
>
> We will clarify and explain this with more details in our revision.

---

> ### Author Response · Authors · 2024-08-10
> **Thanks to Reviewer VV25**
>
> Please allow us to thank you again for reviewing our paper and the valuable feedback, and in particular for recognizing the strengths of our paper in terms of *novel and interesting research problem*, *interesting observation*, *method flexibility*, and *extensive and solid experiments*.
>
> Kindly let us know if our response and the new experiments have properly addressed your concerns. We are more than happy to answer any additional questions during the post-rebuttal discussion period. Your feedback will be greatly appreciated.

---

> ### Author Response · Authors · 2024-08-12
> **A Gentle Reminder of the Post-rebuttal Discussion**
>
> Dear Reviewer VV25,
>
> We would like to sincerely thank you for your helpful comments. We hope our response has adequately addressed your concerns. We take this as a great opportunity to improve our work. We would be very grateful if you could kindly give any feedback to our rebuttal :)
>
> Best Regard,
>
> Paper4797 Author(s)

---

> > ### Comment · Reviewer_VV25 · 2024-08-12
> >
> > I would like to thank the authors for their detailed response and clarifications. The response has addressed most of the concerns, including ZeroMark's applicability to other model architectures and a few confusions in the manuscript.
> >
> > However, there is still one slight concern with regard to ZeroMark's generalizability to discrete data formats (e.g., texts). While one can indeed construct boundary samples using existing adversarial attack methods, from what this reviewer understands, constructing the boundary sample marks only the first step for ZeroMark, after which one still needs to obtain boundary gradients and compute the cosine similarity between the boundary gradient and the watermark pattern. However, for discrete data, the input lies in a discrete space, and thus taking/estimating the gradient w.r.t. the input might not always be feasible. Additionally, for discrete data, the trigger pattern is usually also discrete (e.g., a fixed trigger token or a certain text structure/style), and hence computing cosine similarity could also be infeasible.
> >
> > This reviewer understands that due to page budget and time limitation, it could be difficult to conduct extra evaluations. Nonetheless, this reviewer is still concerned that ZeroMark might not readily applicable to discrete data formats.

---

> > > ### Author Response · Authors · 2024-08-12
> > > **Thank You for Your Positive Feedback and Follow-up Response**
> > >
> > > Dear Reviewer s8Mh,
> > >
> > > We sincerely thank you for your timely follow-up feedback! We are so glad that our previous responses clarified potential misunderstandings and alleviated most of your concerns. We also deeply thank you for your positive feedback, which encourages us a lot! We promise to add more discussions and experiments that we previously committed to in the revision.
> > >
> > > In this response letter, we hope to further alleviate your remain concerns. We believe this can further improve our work. Thank you again for giving us this chance :)
> > >
> > >
> > > ---
> > >
> > > **Q1**: However, there is still one slight concern with regard to ZeroMark's generalizability to discrete data formats (e.g., texts). While one can indeed construct boundary samples using existing adversarial attack methods, from what this reviewer understands, constructing the boundary sample marks only the first step for ZeroMark, after which one still needs to obtain boundary gradients and compute the cosine similarity between the boundary gradient and the watermark pattern. However, for discrete data, the input lies in a discrete space, and thus taking/estimating the gradient w.r.t. the input might not always be feasible. Additionally, for discrete data, the trigger pattern is usually also discrete (e.g., a fixed trigger token or a certain text structure/style), and hence computing cosine similarity could also be infeasible. This reviewer understands that due to page budget and time limitation, it could be difficult to conduct extra evaluations. Nonetheless, this reviewer is still concerned that ZeroMark might not readily applicable to discrete data formats.
> > >
> > > **R1**: Thank you for these insightful comments!
> > >
> > > - **We admit that our ZeroMark is not currently fully ready for discrete data**. As you agree, it's unlikely that we'll be able to fully address this issue in this work not to mention during this rebuttal.
> > > - Nonetheless, prompted and further inspired by your detailed and constructive comments, **we hereby discuss how we can possibly extend our work to discrete data formats** (e.g., texts). We believe this provides a solid foundation for our subsequent and follow-up works.
> > >     - **Estimating the gradients w.r.t. the input**. The estimation of gradients is central to obtaining adversarial examples under the black-box setting. Adversarial examples are obtained by optimizing multiple rounds of estimated gradients based on the original samples. As such, **a well-performing black-box adversarial attack usually means that relatively more accurate gradient estimates can be obtained**. Besides, we can also use a pre-trained word embedding to approximately estimate the gradients. Of course, we admit that this is still only an approximation and an estimate, and its process is subject to some inaccuracies.
> > >     - **Calculating the similarity between gradients and triggers.** We admit that we cannot directly calculate cosine similarity if the trigger pattern is also discrete. Thank you for the reminder! However, we argue that we still have some potential methods to alleviate it.
> > >         - **Using an alternative similarity measure** since gradients and triggers should have some correlations.
> > >         - **Using a meta-classifier to learn these correlations** if we cannot find a proper surrogate measure.
> > >         - **Using a pre-trained word embedding to approximately transfer this discrete optimization to continuous optimization**.
> > >
> > >
> > > We will add more details and discussions in the appendix of our revision.
> > >
> > > ---

---

> > > > ### Comment · Reviewer_VV25 · 2024-08-12
> > > >
> > > > The reviewer would like to thank the authors for the timely response. It has addressed the previous concern. Overall, the paper addresses an interesting topic and the evaluation on image datasets have been quite comprehensive. However, the reviewer decides to maintain the original score, primarily due to the following reason.
> > > >
> > > > The proposed method lacks a theoretical guarantee on the privacy/secrecy of the trigger pattern. ZeroMark proposes to use boundary samples so that the actual trigger pattern is not directly exposed. However, this mainly relies on a heuristic that hiding the actual trigger pattern would hinder the adversary from reconstructing it. This is empirically verified with adaptive attacks, but a theoretical guarantee would have been more persuasive, especially for privacy-preserving tasks. (Note that Theorem 1 in the paper only proves the cosine similarity between the trigger pattern and boundary sample is related; it does not guarantee that it is difficult/impossible for an adversary to reconstruct the trigger.)

---

> > > > > ### Author Response · Authors · 2024-08-12
> > > > >
> > > > > Dear Reviewer VV25,
> > > > >
> > > > > We sincerely thank you for your timely follow-up feedback! We are so glad that our previous responses clarified potential misunderstandings and addressed your concerns. We also deeply thank you for your positive feedback, especially comments regarding our interesting topic and comprehensive evaluation. It encourages us a lot! We promise to add more discussions and experiments that we previously committed to in the revision.
> > > > >
> > > > > We also fully understand your concern and your decision to keep the current score. Empirically, it is difficult for an attacker to recover the watermark patterns from our boundary samples because they contain limited trigger information, measured by mutual information, etc. The information used throughout the optimization process (i.e., the benign samples and the model) is also known to the attacker in advance. However, we do agree that a theoretical guarantee would have been more persuasive, yet we failed to provide it (as we admitted in our limitation analysis). We will further analyze it in our future work.
> > > > >
> > > > > Thank you again for your feedback. Your valuable time and constructive comments are highly important to us!

---

### Author Rebuttal · Authors · 2024-08-07

We thank all reviewers for their constructive feedback, and we have responded to each reviewer individually. We have also uploaded a Rebuttal PDF that includes:

- **Table 1**: The verification performance of ZeroMark for Tiny-ImageNet with SwinTransformer.
- **Tabel 2**: Verification performance averaged over 4 models.

- **Tabel 3-4**: Comparison results averaged over 4 models.

- **Figure 1**: Response to RHDz

- **Figure 2**: Results for ZeroMark against watermark construction attack for BadNets, Blended, WaNet.

- **Tabel 5-6**: The performance of ZeroMark on Blended with ’Hello Kitty'.

---

### Decision · Program_Chairs · 2024-09-25

**Decision:**

Accept (poster)

**Comment:**

The paper receives mostly positive comments on the importance of the addressed task and the novelty of the approach. One reviewer (s8Mh) raised several concerns and the authors discuss with the reviewers in multiple rounds by clarifying most of the reviewer's concerns and the reviewer did not follow up the final comments, which are quite important for the final decision. Reading all review comments and authors' follow-ups, the AC recommends to accept the submission. Please revise the manuscript by the all reviewers comments for the camera ready.